# Contextual cueing of visual search reflects the acquisition of an optimal, one-for-all oculomotor scanning strategy

Werner Seitz [1]✉, Artyom Zinchenko[1], Hermann J. Müller[1,2] & Thomas Geyer[1,2,3]

Visual search improves when a target is encountered repeatedly at a fixed location within a stable distractor arrangement (spatial context), compared to non-repeated contexts. The standard account attributes this contextual-cueing effect to the acquisition of display-specific long-term memories, which, when activated by the current display, cue attention to the target location. Here we present an alternative, procedural-optimization account, according to which contextual facilitation arises from the acquisition of generic oculomotor scanning strategies, optimized with respect to the entire set of displays, with frequently searched displays accruing greater weight in the optimization process. To decide between these alternatives, we examined measures of the similarity, across time-on-task, of the spatio-temporal sequences of fixations through repeated and non-repeated displays. We found scanpath similarity to increase generally with learning, but more for repeated versus non-repeated displays. This pattern contradicts display-specific guidance, but supports one-for-all scanpath optimization.

[1] Department Psychologie, Ludwig-Maximilians-Universität München, Munich, Germany. [2] Munich Center for Neurosciences – Brain & Mind, Ludwig-Maximilians-Universität München, Planegg-Martinsried, Munich, Germany. [3] NICUM - NeuroImaging Core Unit Munich, Ludwig-Maximilians-Universität München, Munich, Germany. ✉email: Werner.Seitz@campus.lmu.de

Visual search for a target object among nontarget, or distractor, objects can be facilitated by prior knowledge of the scene, including contextual long-term memory of co-occurring objects or the position of the target relative to an invariant (spatial) distractor arrangement (e.g., ref. [1]). Chun and Jiang[1], in their seminal study, presented participants with search arrays containing a target letter "T" among a set of distractor letters "L" (a relatively hard search task that affords little bottom-up or top-down guidance; cf[2].). Critically, in half of the trials, the spatial arrangements of the distractor and target stimuli were repeated (i.e., repeated, old contexts); in the other half, the distractor locations were generated anew on each trial, while keeping the target position constant (i.e., nonrepeated, new contexts). Thus, given that the absolute target positions were fixed in both types of trial, the only difference between them was whether or not the target location was predictable from the distractor context. The two sets of repeated and nonrepeated displays were presented randomly intermixed within each block of trials. Chun and Jiang[1] observed that the search reaction times (RTs) decreased with increasing block number for nonrepeated as well as repeated arrays, attributed to general procedural learning of how to perform the task. Critically, however, this practice-dependent improvement was larger for the repeated arrays – an effect Chun and Jiang[1] referred to as contextual cueing (CC). Search RT facilitation by repeated contexts typically emerges rapidly, after just a few (2–4) encounters of the same visual arrangement (e.g., ref. [3]), and it appears to be implicit (and automatic) in nature[4]. Further, in terms of oculomotor search performance, eye-tracking studies indicate that contextual learning leads to a reduction in the number of fixations required to reach the target in repeated, compared to nonrepeated, displays (e.g., ref. [5–9]).

One intriguing, and, as we will argue below, open question is how these savings in the number of fixations are actually produced in repeated search displays. The standard account attributes these savings to search being cued, or guided, more directly to the target location as a result of having acquired a (long-term) associative memory representation, or template, of a specific distractor-target arrangement. This template is activated upon re-encountering such an arrangement on a given trial, which then top-down increases the attentional priority of the target location (e.g., ref. [1]; for computationally explicit models, see, e.g., ref. [10,11]) – thus enhancing the target's potential to summon covert or overt attention. According to this account, the number of attention shifts required to detect a target in a repeated search array will decrease with increasing (re-)encounters of this array, due to the build-up of a search-guiding contextual memory template for this array (e.g., ref. [1]). Support for this comes from studies of contextual cueing that used fixation number as a dependent measure (e.g., ref. [5,6],). These studies showed that finding the target in repeated arrays requires overall fewer eye movements – though with the guidance effect emerging only after the first few fixations, suggesting that it may take some time for the template to come into play. In Tseng and Li's[6] terms, search may involve some 'inefficient', unguided scanning of the array until an informative constellation of distractors represented in the template is encountered. Activation of the template would then lead an effective, guided search phase: a relatively direct homing in of attention on the target location after a series of more exploratory fixations. Accordingly, the savings in the number of eye movements for repeated (vs. nonrepeated) displays would arise from later fixations in the saccadic scanpaths – perhaps with the template-based priority signal pointing to the target location growing increasingly stronger as oculomotor scanning approaches the target item[1]. This specific-template account is attractive, not least because it ties in seamlessly with the functional architecture assumed by general theories of search guidance, such as Guided Search[12–14].

However, there may be an alternative, more procedural account of contextual facilitation that does not rely on the notion that observers acquire memory representations that are specific to particular distractor-target arrangements – a conceptually new account that the present study set out to explore. In fact, procedural learning in CC paradigms is a universal finding in virtually all pertinent studies (for reviews see, e.g., ref. [15,16],): search speed improves, typically quite substantially, over the course of practice on the task, that is, across trial blocks[4]. Importantly, an improvement is evident for nonrepeated – as well as repeated – displays, which is generally attributed to procedural learning, which optimizes, or automatizes, performance through the development/refinement of a task-appropriate (search) settings, akin to the development of a skill (e.g., ref. [17–20],). Critically, though, the improvement is more marked for repeated (vs. nonrepeated) displays, which constitutes the contextual-facilitation effect.

Of note, the extant studies of contextual cueing have almost all examined the facilitation effect (in terms of RT, fixation/saccade number, etc. measures) across aggregated sets of repeated vs. nonrepeated displays. Accordingly, arguably little is known about how contextual facilitation comes about at the single-display level: Is it based on attentional guidance by specific LTM templates of spatial target-distractor relations in individual displays? Or is it due to the acquisition of more display-generic (i.e., relatively display-independent) scanning procedures that are mainly shaped by – and so best adjusted to – the set of repeated displays? On the latter hypothesis, what is optimized in procedural task learning may be a search strategy which is increasingly generic in the sense that it is applicable to all search displays, repeated and non-repeated (rather than being specific for particular repeated displays). However, as a result of *statistical learning*, this strategy is more tuned to, and so more effective for, those displays that are encountered frequently (repeated displays), rather than displays searched only once (nonrepeated displays). Thus, the procedural-learning hypothesis would provide a *unitary* account in that it explains both the general and the specific gains in terms of tuning and optimization of the oculomotor scanning strategy to the regularities prevailing in the whole set of displays that observers encounter over the task. Please note that we evoke a view of optimization according to which visual search is adjusted toward a specific goal, namely, finding and responding to a target letter T in a cluttered array of distractor letters L. Given that this target differs from the distractors only in the combination of two shape features and the T vs. L junction, the search as such is likely inefficient, in terms of producing relatively steep slopes of the function relating RTs to the number of elements in the search display (e.g., ref. [14].). Nevertheless, through procedural learning, performance is optimized to achieve the goal reliably with a minimum of effort.

In sum, in hard search tasks requiring serial eye movements to find the target, repeatedly scanning identically composed item arrays leads to a decrease of RTs and fixation numbers compared to novel displays. However, the eye-movement savings (likely the main driver of the RT savings) occur only relatively late during the trial; and even after a reasonable amount of display repetitions, a considerable number of (some 4–6) fixations is still needed for the eye to reach the target (cf[6].). This suggests that contextual learning may foremostly aid, or optimize, the selection of fixation locations along (at least parts of) the oculomotor scanpath, thereby increasing the likelihood of hitting the target location relatively early during the search. In other words, contextual learning may drive adaptations of participants' general scanning strategies that broadly structure their search in a

display-generic manner that is adapted to repeatedly encountered displays collectively, rather than individually (and that is little influenced by nonrepeated displays, which – by virtue of being encountered only once – cannot consistently contribute to shaping this strategy). Of course, such display-generic learning may operate alongside display-specific learning of the spatial target-distractor relations in individual repeated displays. Arguably, however, relatively direct, display-specific guidance of attention and the eye to the target location may only play limited role at least in hard search tasks requiring serial scanning.

To test this alternative, proceduralization account, we set out to, first of all, establish (and thus replicate) contextual facilitation in terms of the standard summary RT and eye-movement measures that have informed theorizing in the extant contextual-cueing literature. Then, we went on to examine oculomotor-scanpath-similarity measures – in particular, Dynamic Time Warping, Discrete Fréchet Distance, and Area Between Curves – that are diagnostic of similarity in the spatio-temporal sequence of fixations across individual (repeated and nonrepeated) displays, as well as the sequences produced by individual participants. These analyses were designed to reveal detailed information about the proceduralization of search performance, which is lost in the standard averaging of dependent measures both across individual repeated and, respectively, nonrepeated displays and across individual observers.

According to the procedural-optimization hypothesis, (1) scanpaths should become more homogeneous for individual displays across participants over trial blocks, with scanpaths for repeated displays becoming more similar compared to those for nonrepeated displays. Given that (any acquired) display-specific contextual-memory templates take time to become (fully) activated to provide direct guidance (e.g.[6,5,]), higher scanpath homogeneity would particularly reflect display-generic eye-fixation sequences during the earlier, unguided parts of the search. (2) There should be an increased similarity when scanpaths for different displays are compared within individual participants: similarity measures should be higher for pairs of (differently composed) repeated displays compared to pairs of (different) nonrepeated displays. In contrast, the display-specific hypothesis of contextual-cueing would predict that scanpaths become more dissimilar for pairs of repeated (relative to nonrepeated) displays.

## Methods

**Participants**. The sample size was determined based on Vadillo et al.'s[21] meta-analytical study of contextual cueing (i.e., which reported a rather large effect-size score of Cohen's d = 1.00). A power analysis based on this meta-analysis indicated that to find a main effect of contextual cueing on RT performance with 85% power, a minimum sample size of N = 11 participants would be needed. Based on this estimate, when analyzing contextual-facilitation effects at the level of each of our 4 individual repeated displays, this would require at least a 4 times larger sample size. In fact, a sample size of $N > 40$ participants is comparable with other, relevant studies of contextual cueing that have examined contextual facilitation at the level of individual learning blocks/epochs (e.g.[22,]) or of single displays (e.g.[23,24,]). Based on these considerations, we recruited N = 46 participants for the present experiment (38 identifying themselves as female, the remaining 8 as male; 3 left-handed; mean age = 23.28 [SD = 5.62, range = 19–43] years; no data on ethnic identity was collected). Written consent was obtained from each participant; with an ethics approval by the German Research Council (DFG; under GE 1889/4-2). Note that for nonsignificant effects, we additionally report Bayes statistics, where we used the Bayesian Information Criterion as approximation to the Bayes factor (BF$_{10}$; see ref. [25,26]).

**General approach**. Our goal was to bring together established RT and oculomotor measures of the contextual-cueing effect, which focus on group mean values, with oculomotor-scanpath measures that quantitatively describe search behavior in a more fine-grained manner, in particular, at the levels of individual displays or individual participants (see Fig. 1A and B). Additionally, by replicating established measures from the literature (ref. [5,6,27,28]), we aimed to ensure the representativeness of our own data for contextual-cueing studies at large, thus increasing the confidence in the generality of our analyses and findings. Please note that our study was not preregistered as we had a particular – exploratory – focus that seeks to find primary evidence for an alternative, procedural, account of statistical learning in search tasks, by also demonstrating the applicability and potential of scanpath comparison techniques to visual search in repeated versus non-repeated target-distractor arrays and thus generating ideas that justify further research.

For our analysis approach to be feasible, we adjusted the experimental design in two respects: First, and motivated by a previous study of contextual cueing[29], we reduced the number of learnable, repeated target locations, as well as the number of target locations in nonlearnable, nonrepeated displays to four each, with one target location per display quadrant; this was meant to ensure that the memory signals for the respective target location and the corresponding (possibly display-specific) scanpath would have as little interference from other repeated displays as possible and that allocation of attention over space and time would be maximally different. Second, we presented the same repeated and non-repeated display arrangements to all participants, in the same trial order. Using the same set of displays allowed us to control the perceptual content of the display set throughout the experiment; in particular, using the same arrangements for non-repeated displays ensured a "fair" comparison between scanpaths, eliminating confounds originating from, across participants, variably composed distractor-target configurations in non-repeated displays. Methodologically, these adjustments made it possible to compare pairs of scanpaths at different levels and relating to (1) the similarity of fixation sequences through an individual display when viewed by pairs of different participants and (2) the similarity of scanpaths for an individual participant viewing (pairs of) different displays. These design measures enabled us to perform a thorough test of the contrasting predictions made by the specific and the generic procedural-optimization accounts.

We acknowledge that the number of (4) consistently arranged target-distractor displays employed here is relatively low compared to the 8–12 repeated arrays typically used in the relevant studies (see, e.g., ref. [21,]). Assuming that having to deal with fewer repeated displays fosters the acquisition of contextual regularities, the facilitation effect generated under the present conditions may turn out more robust than the meta-analytical effect reported by Vadillo et al.[21], with a Cohen's d effect-size score of 1.00. However, this is not supported by the present RT data (see below), which revealed a Cohen's d = 0.90 (95% CI: 0.30 − 1.51) – rendering it unlikely that contextual facilitation is a simple function of the number of different repeated displays encountered in an experiment. Nevertheless, our participants were presented with identical sets of repeated (and nonrepeated) displays. While this was a necessity for our scanpath analysis to work (in particular, for permitting scanpaths to be compared between different participants searching the same displays), it remains a possibility that the results are bound to these displays. To address this, we used linear mixed models, in which we explicitly took into account the *random* variability coming from individual nonrepeated displays (as well as individual participants) when estimating the effects of our fixed factors of context

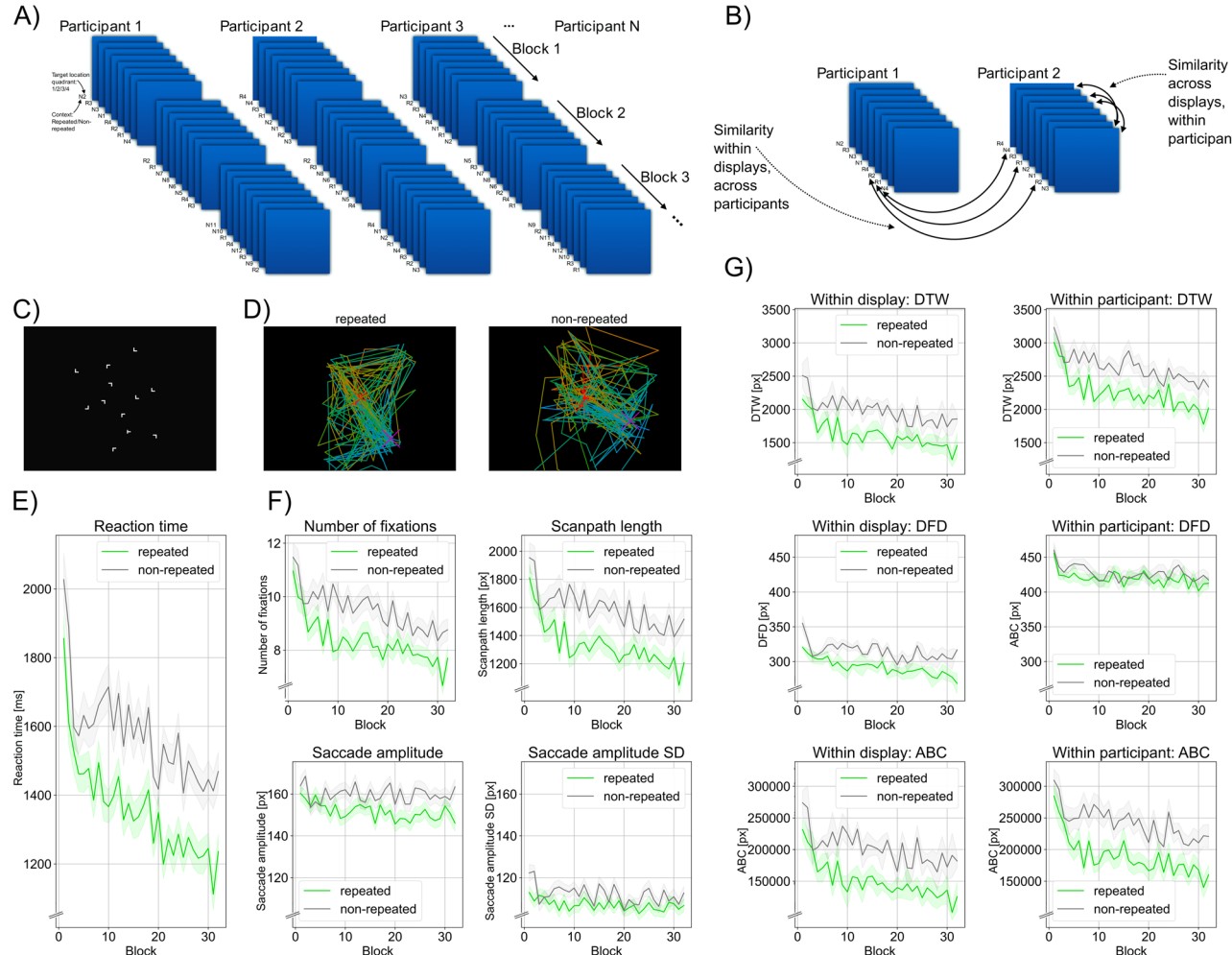

**Fig. 1 Design, analytic approach and results from the current study A** and **B** illustrate the trial schedule: Each of our $N = 46$ participants viewed the same set of (repeated and nonrepeated) displays, allowing us to compare the scanpath similarities within displays and within participants. **C** shows an example display used in the search task, with a right-oriented T target letter located in the lower right quadrant; and (**D**) illustrates participants' scanpaths when searching for Ts (in displays with the target position in the lower-right quadrant marked by a violet cross) in repeated and nonrepeated displays. **E** shows the reaction-time benefit for repeated over nonrepeated displays (in ms) as a function of block; and (**F**) outlines more effective processing in repeated vs. nonrepeated displays when oculomotor parameters are examined: number of fixations, scanpath length, and means and standard deviations of saccadic amplitudes. The data (except for fixation number) represent pixel values: 52 pixels correspond to 1 degree of visual angle. **G** shows blockwise differences between repeated and nonrepeated displays in scanpath similarity measures when computed using the metrics of Dynamic Time Warping, Fréchet Distance, and Area Between Curves, both within displays and within participants. The data are given in pixel values (or, respectively, squared pixel values for ABC). The shaded areas in (**E**–**G**) depict the standard error of the mean.

and block (the specific structure of our models is presented in the Supplementary Methods).

**Apparatus and stimuli**. The experimental routine was programmed in Matlab with Psychtoolbox extensions (ref., [30,31]) and run on an Intel PC under the Windows-7 operating system. Participants were seated in a dimly lit laboratory booth in front of a 19-inch CRT monitor (AOC, Amsterdam; display resolution 1024 × 768 pixels; refresh rate: 85 Hz) at a viewing distance of 60 cm (controlled by a chin rest). The search displays consisted of 12 gray items (luminance: 1.0 cd/m², 1 target and 11 distractors) presented against a black background (0.11 cd/m²). All stimuli extended 0.35° of visual angle in both width and height. As depicted in Fig. 1C, the items were arranged on three (invisible) concentric circles around the display center (with radii of 91, 182, and, 273 pixels for circles 1, 2, and 3, respectively). In *repeated* displays, the locations and orientations of the distractors were held constant across trials; in *non-repeated* displays, all distractors

(i.e., their locations and orientations) were generated anew on each trial. Note that in all presented displays, the location of the target was repeated but the (left/right) orientation of the target was determined randomly and was, thus, unpredictable. As a result, a repeated context could only be associated with a specific (repeated) target location, but not with a specific target identity. Following Chun and Jiang[1], this approach is used in most CC studies to ensure that contextual facilitation of RTs is owing to the repeated context guiding attention/the eyes, rather than facilitating the selection of the manual response (invariably) associated with a given repeated display. Importantly, both the set of $N = 4$ repeated displays and the set of $N = 128$ randomly generated nonrepeated displays were kept constant across all 46 participants, so that each participant encountered identical repeated and non-repeated configurations. Note, however, that trial order was randomly chosen within each block of $N = 4$ repeated plus $N = 4$ nonrepeated trials for individual participants. This enabled us to keep low-level, individual display properties

constant across participants and thus compute dependent – scanpath – measures for each individual display (with variations between participants providing the error term).

There were overall 8 possible target locations, 4 of which were used for *repeated* displays (with constant distractor layouts) and the other 4 for *non-repeated* displays (with random distractor arrangements). Keeping the target locations constant in non-repeated as well as repeated displays is a standard procedure in studies of contextual cueing, to permit the learning of invariant target-distractor contexts to be dissociated from absolute target-position learning (i.e., target-location probability cueing; e.g., ref. [32,33],): absolute target-location learning is effectively controlled for by maintaining constant target positions in both repeated and nonrepeated displays. All targets, in both types of display, were located on the second ring, controlling for the distance of the target from to the display center in all conditions. Furthermore, the targets were placed in all four quadrants with equal probability. Specifically, the (12) display items were randomly assigned to 12 out of a total of 40 possible locations (4, 8, 12, and 16 locations on ring 1, 2, 3, and 4, respectively), the only constraint being that each quadrant contained an equal number of items (either 3 distractors or 1 target and 2 distractors). This means that in principle, there were more than $2.07*10^9$ variants for generating individually unique search displays. Selecting our repeated (and non-repeated) displays randomly from this large space of possible configurations rendered it quite unlikely that they were structurally highly similar.

Importantly, participants were not informed about the fact that some of the search arrays were presented repeatedly. The "T" target was rotated randomly by 90° to either the left or the right. The 11 remaining items were L-shaped distractors rotated randomly at orthogonal orientations (0°, 90°, 180°, or 270°). Figure 1C presents example display layouts (see also Supplementary Fig. 1) To record eye movements, a video-based eye-tracker was used (EyeLink 1000; SR Research Ltd., Mississauga, Ontario, Canada; version 4.594), monitoring participants' right eye. A standard 9-point (grid) calibration of the eye tracker was performed initially and validated at the beginning of each fourth block of (32) trials. Calibration was considered accurate when fixation positions fell within ~1.0° (=diameter) of visual angle for all calibration points. The average calibration error was 0.52°, the average of the maximal errors was 0.86. No data had to be removed due to poor calibration. Calibration accuracy was further checked by the experimenter on each trial. The default psychophysical sample configuration of the eye-tracking system (i.e., saccade velocity threshold set at 35°/s, saccade acceleration threshold set at 9500°/s²) was adopted for identifying saccadic eye movements.

**Trial sequence**. A trial started with the presentation of a central fixation cross (0.10° x 0.10°, luminance: 1.0 cd/m2) for 500 ms. Next, the fixation cross was removed from the screen, and, following a blank interval of 200 ms, the search display was presented. Observers were instructed to find the target "T" and respond as quickly and accurately as possible to its (left vs. right) orientation, while being allowed to move their eyes freely. Each search display stayed on the screen until a manual response was elicited. If the "T" was rotated to the right (left), observers responded by pressing the right (left) arrow button on a computer keyboard with their right (left) index finger. Following a response error, the word "Wrong" appeared in the screen center for 1000 ms. Each trial was followed by a blank inter-trial interval of 1000 ms. The experiment consisted of 256 trials (32 blocks x 8 trials each, 50% repeated displays in each block). Participants

were free to proceed to the next block at their own pace. The search task took some 30 min to complete.

**Recognition test**. At the end of the experiment, observers performed a yes/no (repeated/nonrepeated display) recognition test, permitting us to assess whether they had acquired any explicit memory of repeated configurations presented in the preceding search task (a standard procedure in contextual-cueing experiments; see e.g., ref., [1]). To this end, observers were presented with 4 repeated displays and 4 newly composed displays. The task was to indicate whether or not a given display had been shown previously, by pressing the left or the right mouse button, respectively. The 4 repeated and the 4 newly generated displays were presented in random order. Observers' responses in the recognition task were nonspeeded and no error feedback was provided.

**Statistical analysis**. Comparisons of scanpaths were carried out in Python[34]. Statistical analysis was performed using Python[34], as well as R (version 3.4.3, ref. [35,36]). We analyzed our dependent measures of reaction times, error rates, and oculomotor variables using the lme4 package in R for linear mixed effect modeling, including target quadrant and participant as random factors in addition to the fixed factors of block and context. All tests reported in this study were conducted as two-sided parametric tests and the aptness of these tests was checked by visualization and formal methods.

Previous studies of contextual cueing reported substantial target-quadrant- and participant-dependent variations in baseline RTs (e.g., ref. [37,38],). Accordingly, we used linear random-intercept models (for the numerical dependent variables of RTs, error rates, oculomotor measures) to account for unwanted variability deriving from individual displays (with different target locations/quadrants) and individual participants. Also, by considering target quadrant and participant as random factors, we ensured that our results would be as generalizable as possible to other studies of contextual cueing. Note that the analyses of the scanpath metrics (see below) required specific variability coming from comparisons of individual displays and, respectively, of individual participants. For this reason, target quadrant and participant were not included as random factors in these analyses. Our model fits (i.e., effect sizes) were quantified in terms of Nakagawa's $R^{2,38}$ using the package "performance" in R.

**Dependent measures**. To establish comparability between the present investigation and previous contextual-cueing studies, as well as to validate our dataset as being representative for visual-search paradigms, we begin our analysis with an examination and replication of established measures of the contextual-facilitation effect: RT, fixation number, and saccade amplitude (see, ref. [6], or[5]). We then proceed to the presentation of scanpath measures of contextual facilitation, including the total length of the scanpaths (ref. [27,39]) and the standard deviation of the lengths of the saccades constituting each scanpath. The latter essentially provides a new measure of the variability of saccade lengths across individual observers and displays, where a decrease in variability can be considered a measure of automaticity[40]. This is followed by overlay-plot visualizations of individual participants' spatio-temporal sequence of oculomotor behavior (which are also meant to demonstrate the usefulness of our scanpath approach to eye-tracking investigations of visual search in general). From these visualizations, a quantitative measure of contextual cueing, namely: scanpath similarity or consistency, is derived.

**Analysis of scanpath similarity**. The similarity of scanpaths was computed using established measures in the field (cf[41,42].), in

particular, Dynamic Time Warping, Discrete Fréchet Distance, and Area Between Curves.

Dynamic Time Warping is a measure of similarity between two fixational series of different lengths. Two individual scanpaths may be highly similar with regard to the placing (i.e., the spatial coordinates) of individual fixations, but the temporal alignment of these sequences may be less consistent across individual trials. The strength of Dynamic Time Warping is that it can quantify the similarity of the shapes of scanpaths with distinct time series. Specifically, this metric compares two fixational series by aligning them in the time domain, thus minimizing the Euclidean distance between the aligned series. Concerning the Discrete Fréchet Distance: this metric can also deal with fixational time series of different length (and tempo). The Fréchet Distance considers both the location and ordering of the individual fixations along two scanpath curves and can be defined with regard to an analogy: a person that is walking a dog on a leash, with the person walking on one (scan-) path/curve and the dog on another path/curve. The discrete version of the Fréchet Distance only compares distances between fixations and not points in between. Hence, the Discrete Fréchet Distance corresponds to the length of the shortest leash possible for traversing both curves. Finally, we computed scanpath similarity based on Area Between Curves, which, like Dynamic Time Warping and the Discrete Fréchet Distance, permits comparisons of scanpaths with different lengths, although the particular scanpath measure is based on the area that falls between two scanpath curves. As Area Between Curves is well-suited to quantify hysteresis[43], this measure should be particularly sensitive to capture scanpath similarity when trajectories have the same start and end points (initial fixation point and target location).

We chose to explore three scanpath metrics, rather than just one, in order to provide a maximally precise and unbiased measurement of the effects of search task training on oculomotor behavior (as there is not yet a consensus which one of the scanpath measures is to be preferred over the others[41]. Our specific trial schedule (see Fig. 1A and B) allowed us to examine the similarity of scanpaths in multiple ways (see Supplementary Table). First, we compared each possible pair of gaze patterns arising from identical displays over different participants. This approach enabled us to compute scanpath similarity for each experimental block (each consisting of 4 repeated and 4 nonrepeated arrays), thereby addressing the important question of how the consistency of viewing patterns changes as a function of practice on the task. Second, we computed similarity of oculomotor trajectories between each pair of different displays viewed by the same participant. This analysis was intended as a strong test of the display-specific vs. general-procedural accounts of contextual repetitions on search-task training.

To more formally examine whether these observations represent meaningful effects, we computed scanpath similarity for each experimental block (1–32). To recap our hypothesis: if participants are learning a generic search procedure that is increasingly effective, then similarity of scanpaths should increase over time for both repeated and nonrepeated arrays, though this effect should be more pronounced for the former displays which, due to being repeated, accrue a greater weight in shaping the generic search procedure. But the prediction would be fundamentally different for the display-specific hypothesis of contextual cueing, according to which experience with individual repeated displays leads to the build-up of display-specific memories and associated scanning behavior. Accordingly, scanpath similarity obtained for pairs of individual repeated displays with different spatial composition should decrease with increased search-task training (and similarity measures should effectively be lower than those for nonrepeated displays). Two analyses were conducted

(see Fig. 1A and B and Supplementary Table). In the first, *within-display* analysis, similarity of eye-movement sequences was calculated from each pair of different participants when viewing a given, individual (repeated or nonrepeated) display. Second, in the within-participant analysis, similarity measures were generated from eye-movement sequences in pairs of different displays when searched by an individual participant. Both analyses were conducted for three similarity measures: Dynamic Time Warping, Discrete Fréchet Distance, and Area Between Curves. Statistical inference was based on linear mixed models with the fixed factors of Context and Block and the random factors of Target Quadrant (in the analysis of within-display similarity) and Participant (within-participant scanpath analysis).

**Validity check of scanpath measures**. Taking the distance of each fixation from the target position into account, search can be divided into an initial inefficient and a subsequent efficient phase (ref. [6]; see also ref. [8,39]): only in the latter does the distance of a given fixation from the target decrease monotonically with each successive fixation. That is, in hard search tasks requiring oculomotor scanning, high-resolution (foveal) vision ultimately ends in the target region. Moreover, these studies found that the RT advantages for repeated over nonrepeated displays were accompanied by fewer fixations in the ineffective, but not the effective, search phase. Based on these observations, we expected higher scanpath similarity – indicative of display-generic scanning procedures – to manifest particularly in the initial 50% of the scanpath trajectories – as compared to the final 50%, as the eye eventually approach the unique target regions in individual repeated and nonrepeated displays (which would lower the within-participant display similarity). A 50/50 comparison of the scanpath-similarity scores confirmed this prediction: when measuring scanpath similarity across all repeated and nonrepeated displays encountered by individual participants, we found similarity (as measured by all three scanpath metrics) to be overall higher in the initial vs. the final scanpath parts: Dynamic Time Warping, DTW, $t(45) = 45.39$, $p < 0.001$, $\eta^2$ (partial) = 0.98, 95% CI: 0.97, 0.99; Discrete Fréchet Distance, DFD, $t(45) = 37.98$, $p < 0.001$, $\eta^2$ (partial) = 0.97, 95% CI: 0.95, 0.98; Area Between Curves, ABC, $t(45) = 33.81$, $p < 0.001$, $\eta^2$ (partial) = 0.96, 95% CI: 0.94, 0.97;. At the same time, the average distance of fixations from the individual-unique target location in repeated and nonrepeated arrays were reliably shorter for the final vs. the initial fixations, $t(45) = 43.37$, $p < 0.001$, $\eta^2$ (partial) = 0.98, 95% CI: 0.96, 0.98. This indicates that scanpath similarity decreases as the eye homes in on physically different target locations in the various displays. Most importantly, our theoretical scanpath and empirical distance measures showed a correlation. To examine this, we calculated a scanpath-similarity difference measure by subtracting the similarity scores in the final from those in the initial scanpath parts; accordingly, higher (i.e., positive) difference values indicate higher similarity in the initial part. Likewise, we computed a difference measure for the average physical distance of initial and final fixations in the scanpath part from the target position (i.e., distance initial fixations minus distance final fixations); accordingly, higher (i.e., positive) values indicate larger target-fixation distances in the first part. These scanpath-difference and fixation-distance measures were computed for each individual participant and then correlated in the complete sample. We found significant positive correlations between the two measures for each scanpath metric, ranging from $r = 0.29$ (ABC, 95% CI: 0.002, 0.54, $p = 0.049$) over $r = 0.37$ (DFD; 95% CI: 0.092, 0.60, $p = 0.0011$) to $r = 0.56$ (DTW, 95% CI: 0.32, 0.73, $p < 0.001$). These findings indicate that our three scanpath-similarity measures are consistently precise in capturing basic

**Table 1 The table presents summary and inference statistics for the four computed oculomotor measures of fixation number, scanpath length, average saccade amplitude, and standard deviation of saccade amplitude (the three latter in pixels).**

**Oculomotor measures**

| Number of fixations | Scanpath length |
|---|---|
| Main effect of Context | Main effect of Context |
| Mean repeated = 8.44 | Mean repeated = 1349.26 |
| Mean nonrepeated = 9.62 | Mean nonrepeated = 1611.78 |
| $F(1, 8701.8) = 136.62$ | $F(1, 8702.3) = 158.04$ |
| $p < 0.001$ | $p < 0.001$ |
| $\eta^2$ (partial) = 0.02 (95% CI: 0.01, 0.02) | $\eta^2$ (partial) = 0.02 (95% CI: 0.01, 0.02) |
| Main effect of Block | Main effect of Block |
| Mean $_{first\ block}$ = 11.29 | Mean $_{first\ block}$ = 1903.37 |
| Mean $_{last\ block}$ = 8.36 | Mean $_{last\ block}$ = 1387.34 |
| $F(31, 8701.7) = 6.73$ | $F(31, 8702.1) = 5.50$ |
| $p < 0.001$ | $p < 0.001$ |
| $\eta^2$ (partial) = 0.02 (95% CI: 0.01, 0.03) | $\eta^2$ (partial) = 0.02 (95% CI: 0.01, 0.02) |
| Interaction Block x Context | Interaction Block x Context |
| $F(31, 8702.3) = 1.04$ | $F(31, 8703.0) = 1.00$ |
| $p = 0.41$ | $p = 0.46$ |
| $\eta^2$ (partial) = $3.69*10^{-3}$ (95% CI: 0.00, 0.00) | $\eta^2$ (partial) = $3.55*10^{-3}$ (95% CI: 0.00, 0.00) |
| BF = $7.74*10^{-55}$ | BF = $4.43*10^{-55}$ |
| Variance explained | Variance explained |
| $R^2_{conditional}$ = 0.13 (95% CI: 0.11, 0.17) | $R^2_{conditional}$ = 0.10 (95% CI: $6.8*10^{-2}$, 0.13) |
| $R^2_{marginal}$ = $3.7*10^{-2}$ (95% CI: $3.7*10^{-2}$, $5.1*10^{-2}$) | $R^2_{marginal}$ = $3.7*10^{-2}$ (95% CI: $3.6*10^{-2}$, $5.2*10^{-2}$) |
| **Average saccade amplitude** | **Standard deviation of saccade amplitude** |
| Main effect of Context | Main effect of Context |
| Mean repeated = 151.72 | Mean repeated = 107.02 |
| Mean nonrepeated = 160.01 | Mean nonrepeated = 112.24 |
| $F(1, 8701.8) = 84.73$ | $F(1, 8702.4) = 42.59$ |
| $p < 0.001$ | $p < 0.001$ |
| $\eta^2$ (partial) = $9.64*10^{-3}$ (95% CI: 0.01, 0.01) | $\eta^2$ (partial) = $4.87*10^{-3}$ (95% CI: 0.00, 0.00) |
| Main effect of Block | Main effect of Block |
| Mean $_{first\ block}$ = 162.55 | Mean $_{first\ block}$ = 118.17 |
| Mean $_{last\ block}$ = 155.63 | Mean $_{last\ block}$ = 109.95 |
| $F(31, 8701.7) = 1.48$ | $F(31, 8702.2) = 1.82$ |
| $p = 0.041$ | $p = 0.0036$ |
| $\eta^2$ (partial) = $5.24*10^{-3}$ (95% CI: 0.00, 0.00) | $\eta^2$ (partial) = $6.44*10^{-3}$ (95% CI: 0.00, 1.00) |
| Interaction Block x Context | Interaction Block x Context |
| $F(31, 8702.3) = 0.97$ | $F(31, 8703.2) = 0.88$ |
| $p = 0.52$ | $p = 0.66$ |
| $\eta^2$ (partial) = $3.44*10^{-3}$ (95% CI: 0.00, 0.00) | $\eta^2$ (partial) = $3.12*10^{-3}$ (95% CI: 0.00, 0.00) |
| BF = $2.52*10^{-55}$ | BF = $6.36*10^{-56}$ |
| Variance explained | Variance explained |
| $R^2_{conditional}$ = 0.12 (95% CI: $9.9*10^{-2}$, 0.17) | $R^2_{conditional}$ = $8.2*10^{-2}$ (95% CI: $6.1*10^{-2}$, 0.12) |
| $R^2_{marginal}$ = $1.6*10^{-2}$ (95% CI: $1.6*10^{-2}$, $2.8*10^{-2}$) | $R^2_{marginal}$ = $1.3*10^{-2}$ (95% CI: $1.4*10^{-2}$, $2.5*10^{-2}$) |

properties of serial visual search: they decrease in the later, efficient parts of the scanpath, as the eyes move nearer the different target regions – providing a validity check (vis-à-vis established effects) for our analysis approach.

**Reporting summary**. Further information on research design is available in the Nature Portfolio Reporting Summary linked to this article.

## Results

To preview our results: (1) we replicate previous findings of more efficient visual search in terms of expedited RTs and reduced fixation number (and other established oculomotor measures) for repeated vs. nonrepeated search arrays. (2) Individual

participants exhibit scanpath patterns that are increasingly similar across blocks of repeated and nonrepeated displays – both within displays and within participants. Repeated displays nevertheless exhibit a higher consistency within displays and within participants, reflected in a higher degree of scanpath similarity across repeated displays.

**Reaction times**. For the RT analyses, error trials and extreme RTs three standard deviations below and above the mean were excluded from the data. This outlier criterion led to the removal of ~3% of all trials. Overall, observers had an average error rate of ~1.5%, without any indication of a speed-accuracy trade-off. Regarding error rates, no main effect (context: $F(1, 2810.2) = 0.13$, $p = 0.72$, $\eta^2$ (partial) = $4.63e*10^{-5}$, 95% CI: 0.00, 0.00, BF = $1.97*10^{-81}$; block: $F(31, 2811.0) = 1.10$, $p = 0.32$, $\eta^2$ (partial) = $3.91e*10^{-4}$, 95% CI: 0.00, 0.00, BF = $9,97*10^{-164}$) nor interaction effect ($F(31,2810.2) = 0.97$, $p = 0.51$, $\eta^2$ (partial) = 0.01, 95% CI: 0.00, 0.01, BF = $5.55*10^{-78}$) reached significance.

The analysis of the mean RTs revealed a main effect of Context: participants responded faster to repeated relative to nonrepeated displays (1359 vs. 1574 ms, $F(1, 8701.6) = 268.0207$, $p < 0.001$, $\eta^2$ (partial) = 0.03, 95% CI: 0.02, 0.04). The main effect of Block was also significant, reflecting a decrease in RTs with increasing block number (block 1 = 1960 ms; block 32 = 1345 ms, $F(31, 8701.5) = 13.00$, $p < 0.001$, $\eta^2$ (partial) = 0.04 (95% CI: 0.03, 0.05). The Context × Block interaction was nonsignificant ($F(31, 8701.9) = 0.88$, $p = 0.67$, $\eta^2$ (partial) = $3.13*10^{-5}$ (95% CI: 0.00, 0.00), BF = $5.86*10^{-56}$), indicative of a stable contextual-facilitation effect across blocks (cf. Figure 1E and Supplementary Fig. 2), corresponding to an overall explained variance of $R^2_{conditional}$ = 0.19 (95% CI: 0.16, 0.23) and, removing the random effects, an $R^2_{marginal}$ = $6.4*10^{-2}$ (95% CI: $5.9*10^{-2}$, $7.5*10^{-2}$), respectively.

**Recognition performance**. Participants' comparison of the hit rates against the 50% baseline – chance – performance yielded a nonsignificant result, $t(45) = 1.09$, $p = 0.28$, $\eta^2$ (partial) = 0.03 (95% CI: 0.00, 0.17), BF = 0.28. Thus, there was little indication of explicit, above-chance recognition of displays that had been encountered repeatedly over the course of the search task.

**Oculomotor performance**. Fig. 1F (see also Supplementary Fig. 3) presents a series of oculomotor measures, derived from fixations and saccades, as a function of block number aggregated over trials, separately for repeated and nonrepeated displays; Table 1 summarizes the respective descriptive and inference statistics when submitting the data to a linear mixed model with Block and Context as fixed factors and Participant and Target Quadrant as random factors. The upper left subplot of Fig. 1F illustrates the second classical finding: a decline in the number of fixations across blocks, with overall fewer fixations made in repeated vs. nonrepeated displays. The upper right plot of Fig. 1F depicts scanpath length. Since target positions were placed at equal distance from the screen center, the scanpath length also coincides with the so-called scan-pattern ratio[27]: the total distance (in pixels) that the eyes traversed across a given search display until arriving at the target location, divided by the shortest distance possible between the initial fixation and the target location – essentially quantifying how directly the eyes approached the target (see, e.g.[7],). This measure turned out significantly smaller for repeated compared to nonrepeated displays. Moreover, as can be seen in the lower left plot of Fig. 1F, repeated configurations showed a smaller mean saccade amplitude. Of note, the average standard deviation of the saccade amplitudes

**Table 2 Summary inference statistics for the two levels of analysis – within display and, respectively, within participant – for the three scanpath similarity measures of Dynamic Time Warping (DTW), Discrete Fréchet Distance (DFD), and Area Between Curves (ABC).**

**Similarity within display**

**DTW**
Main effect of Context
Mean $_{repeated}$ = 1613.82
Mean $_{nonrepeated}$ = 1987.74
$F(1,2760.0)$ = 154.62
$p < 0.001$
$\eta^2$ (partial) = 0.05 (95% CI: 0.04, 0.07)
Main effect of Block
Mean $_{first\ block}$ = 2329.61
Mean $_{last\ block}$ = 1657.38
$F(1,2760.1)$ = 6.48
$p < 0.001$
$\eta^2$ (partial) = $2.34*10^{-3}$ (95% CI: 0.00, 0.01)
Interaction Block x Context
$F(1,2760.1)$ = 1.10
$p = 0.32$
$\eta^2$ (partial) = $3.98*10^{-4}$ (95% CI: 0.00, 0.00)
BF = $4.12*10^{-33}$
Variance explained
$R^2_{conditional}$ = 0.50 (95% CI: 0.46, 0.66)
$R^2_{marginal}$ = 0.40 (95% CI: 0.35, 0.54)

**DFD**
Main effect of Context
Mean $_{repeated}$: 291.7
Mean $_{nonrepeated}$: 314.64
$F(12760.0)$ = 14.90
$p < 0.001$
$\eta^2$ (partial) = $5.37*10^{-3}$ (95% CI: 0.00, 0.01)
Main effect of Block
Mean $_{first\ block}$: 337.8
Mean $_{last\ block}$: 292.78
$F(1,2760.1)$ = 2.11
$p = 0.001$
$\eta^2$ (partial) = $7.64*10^{-4}$ (95% CI: 0.00, 0.01)
Interaction Block x Context
$F(1,2760.0)$ = 0.71
$p = 0.88$
$\eta^2$ (partial) = $2.57*10^{-4}$ (95% CI: 0.00, 0.00)
BF = $1.44*10^{-32}$
Variance explained
$R^2_{conditional}$ = 0.47 (95% CI: 0.46, 0.66)
$R^2_{marginal}$ = 0.38 (95% CI: 0.34, 0.57)

**ABC**
Main effect of Context
Mean $_{repeated}$ = 149860.73
Mean $_{non-repeated}$ = 202522.02
$F(1,2760.0)$ = 193.54
$p < 0.001$
$\eta^2$ (partial) = 0.07 (95% CI: 0.05, 0.08)
Main effect of Block
Mean $_{first\ block}$ = 252733.5
Mean $_{last\ block}$ = 153740.48
$F(1,2760.1)$ = 5.80
$p < 0.001$
$\eta^2$ (partial) = $2.10*10^{-3}$ (95% CI: 0.00, 0.01)
Interaction Block x Context
$F(1,2760.0)$ = 1.06
$p = 0.37$
$\eta^2$ (partial) = $3.84*10^{-4}$ (95% CI: 0.00, 0.00)
BF = $7.64*10^{-33}$
Variance explained
$R^2_{conditional}$ = 0.52 (95% CI: 0.49, 0.69)
$R^2_{marginal}$ = 0.41 (95% CI: 0.33, 0.57)

was also significantly reduced for repeated vs. nonrepeated displays – see lower right plot of Fig. 1f. For all measures, the block x context interaction was not significant (see Table 1).

**Scanpath analysis**. In a first, qualitative analysis, we visualized scanpaths across blocks and target positions. In more detail, we plotted the scanpath representations for each of the four target positions in repeated and, respectively, nonrepeated displays (i.e.,

the four repeated and nonrepeated trials within a block). As can be seen from Fig. 1A (see also Supplementary Fig. 4), in the last block of the search task – after ample opportunity for contextual adaptation – there was a higher degree of similarity between scanpaths for repeated relative to nonrepeated contexts. Specifically, in the repeated condition for each display, saccades tended to be more often and more closely executed in parallel direction across observers, indicative of systematic biases in observers' oculomotor behavior. Also, in the repeated compared to the nonrepeated condition, the color of the lines connecting successive fixation locations (with green denoting the initial and blue the final saccade on a given trial, and intermediate colors denoting saccades in between) tended to be more clustered spatially, indicating that saccades were executed in a more systematic order as well.

**Similarity analysis**. The main finding of this study is that scanpath similarity is increasing throughout the experiments, with repeated displays gaining a significant advantage early on which remains throughout the experiment. These findings are indicative of the convergence towards an optimal search strategy on the level of the set of displays (see discussion). For both analyses, within displays (Table 2) and within participants (Table 3) respectively (also, see Fig. 1G and Supplementary Fig. 5), we found significant main effects of Context and Block, while the Context × Block interactions were nonsignificant. The pattern reflects a steady increase in scanpath similarity (corresponding to smaller numerical values in Dynamic Time Warping, Discrete Fréchet Distance, and Area Between Curves) over the course of the experiment (main effect of Block), which was however higher in repeated vs. nonrepeated displays (main effect of Context) with a stable context effect emerging early on (no significant interaction). Of note, the three similarity measures yielded qualitatively similar results, despite being sensitive to slightly different aspects of the scanpaths, attesting to a high reliability of our findings.

Thus, our scanpath similarity measures provide strong support for a procedural-optimization hypothesis, according to which participants, over time on task, learn a generic search procedure that is increasingly effective for all – repeated and nonrepeated – displays, though repeated displays weigh more highly in the optimization as a result of being searched more often.

**Discussion**

To gain an in-depth understanding of contextual facilitation, we analyzed established measures of the contextual-facilitation effect that focus on aggregate oculomotor indices, as well as new measures based on spatio-temporal scanpath sequences. Concerning individual eye-movement patterns: replicating prior reports, we found that detecting a target involves fewer fixations in repeated compared to nonrepeated target-distractor arrangements (e.g. ref. [5,6,43],), as well as a shorter scanpath length and, accordingly, a smaller scan-pattern ratio (ref. [7,27]). We also found the saccade amplitudes to be significantly shorter for repeated displays (in contrast to[6]). In addition, we established a aggregate oculomotor index of contextual facilitation (that hitherto had not been reported in the literature): a reduced standard deviation of the saccade amplitudes for repeated vs. nonrepeated displays.

**Contextual-cueing of visual search: general procedural guidance of the eyes**. Having established comparability of the present findings with those reported in prior studies of contextual cueing, we went on to examine the oculomotor scanpaths in order to differentiate between a template-based, display-specific and a procedural, display-generic scanning account of acquired contextual facilitation that may drive the gains in the aggregate eye-

**Table 3 Summary inference statistics for the two levels of analysis – within display and, respectively, within participant – for the three scanpath similarity measures of Dynamic Time Warping (DTW), Discrete Fréchet Distance (DFD), and Area Between Curves (ABC).**

**Similarity within participant**

**DTW**
Main effect of Context
Mean $_{repeated}$ = 2247.55
Mean $_{nonrepeated}$ = 2628.75
$F(1,189) = 99.32$
$p < 0.001$
$\eta^2$ (partial) = 0.34 (95% CI: 0.24, 0.44)
Main effect of Block
Mean $_{first\ block}$ = 3119.54
Mean $_{last\ block}$ = 2180.57
$F(31,189) = 2.80$
$p < 0.001$
$\eta^2$ (partial) = 0.31 (95% CI: 0.11, 0.34)
Interaction Block x Context
$F(31,189) = 0.58$
$p = 0.96$
$\eta^2$ (partial) = 0.09 (95% CI: 0.00, 0.01)
$BF = 8.73*10^{-47}$
Variance explained
$R^2_{conditional}$ = 0.34 (95% CI: 0.29, 0.43)
$R^2_{marginal}$ = $8.9*10^{-2}$ (95% CI: $8.5*10^{-2}$, 0.12)

**DFD**
Main effect of Context
Mean $_{repeated}$: 419.14
Mean $_{nonrepeated}$: 427.24
$F(1,189) = 91.22$
$p < 0.001$
$\eta^2$ (partial) = 0.33 (95% CI: 0.22, 0.42)
Main effect of Block
Mean $_{first\ block}$: 458.01
Mean $_{last\ block}$: 415.03
$F(31,189) = 2.84$
$p < 0.001$
$\eta^2$ (partial) = 0.32 (95% CI: 0.12, 0.35)
Interaction Block x Context
$F(31,189) = 0.64$
$p = 0.93$
$\eta^2$ (partial) = 0.10 (95% CI: 0.00, 0.03)
$BF = 1.81*10^{-49}$
Variance explained
$R^2_{conditional}$ = 0.28 (95% CI: 0.21, 0.36)
$R^2_{marginal}$ = $2.6*10^{-2}$ (95% CI: $3.2*10^{-2}$, $5.2*10^{-2}$)

**ABC**
Main effect of Context
Mean $_{repeated}$: 188,744.43
Mean $_{nonrepeated}$: 242078.25
$F(1,189) = 107.64$
$p < 0.001$
$\eta^2$ (partial) = 0.36 (95% CI: 0.26, 0.46)
Main effect of Block
Mean $_{first\ block}$: 297143.1
Mean $_{last\ block}$: 190412.32
$F(31,189) = 2.87$
$p < 0.001$
$\eta^2$ (partial) = 0.01 (95% CI: 0.00, 0.07)
Interaction Block x Context
$F(31,189) = 0.61$
$p = 0.95$
$\eta^2$ (partial) = $3.22*10^{-3}$ (95% CI: 0.00, 0.04)
$BF = 4.72*10^{-47}$
Variance explained
$R^2_{conditional}$ = 0.30 (95% CI: 0.26, 0.37)
$R^2_{marginal}$ = 0.10 (95% CI: $0.2*10^{-2}$, 0.13)

movement indices (such as the reduced total fixation number) for repeated displays. Using two distinct approaches – of comparing eye movement sequences between pairs of identical displays when viewed by different participants, and, respectively, individual participants viewing different displays – and three metrics of scanpath similarity (Dynamic Time Warping, Discrete Fréchet Distance, and Area Between Curves), we found that, while the consistency of the scanpaths increased overall with increasing

time on task for both repeated and nonrepeated displays, these practice-dependent gains were more strongly driven by displays sampled repeatedly (vs. displays sampled only once). This effect pattern is indicative of common regularities shared between the (statistical) sample of search displays, which influence scanpaths in a way that is independent of the particular arrangement encountered on a trial or even the individual participant – akin to generic, one-for-all procedural learning. Had there been display-specific learning of individual repeated displays, the similarity measures obtained from any two such displays should have been reduced, relative to nonrepeated (baseline) displays, as each repeated array should have come to elicit its unique scanning pattern. However, at variance with this prediction from display-specific learning accounts of contextual cueing, we found the similarity measures to be actually different.

Of note, this pattern of scanpath-similarity effects does not rule out that the scanpaths become tuned to specific displays at some point along their progression, for instance, when the eye finally homes in on the target location;[6] nor do we take this to rule out the possibility of display-specific learning under all circumstances (considered further below). Rather, we take our findings to demonstrate that, over the course of a hard search task, the notion of display-general learning provides an apt account of contextual facilitation. In line with this[6]; (see also ref. [8,39]) have proposed that the (efficient) phase, in which the eyes come closer to the target with each successive fixation, does not differ significantly between repeated and nonrepeated displays; but the two display types differ with respect to the number of fixations in the preceding inefficient phase (with reduced fixation numbers in repeated displays). However, in contrast to Tseng and Li[6], our scanpath-similarity analyses, which take into account the entire spatio-temporal series of fixation events, show that eye movements in the so-called ineffective phase are, in fact, not (in terms of Tseng & Li[6] p. 1371) "wasted". Instead, our findings of increasing scanpath homogeneity with extended time on task suggest that what may look an ineffective phase actually constitutes an important period during which procedural learning of a general scanning scheme becomes functional.

In terms of skill acquisition[18], when participants perform a new task of searching displays with novel, as yet to-be-discovered statistical properties, one would expect that they learn to adapt and optimize their oculomotor scanning behavior with respect to the display statistics in rather generic terms – as opposed to acquiring search-guiding memory representations tailored to specific, individual display arrangements, as assumed by standard accounts of contextual cueing. These scanning strategies are optimal in the sense that they save cognitive effort: they do not require memorizing arbitrary distractor-target configurations and expensively checking a given display arrangement against a set of representations in contextual memory. As an unavoidable side effect, developing a strategy that is optimally adjusted to the statistical search environment at large (with repeated displays having a greater weight in shaping this strategy due to their increased frequency of occurrence) would also optimize the scanning of nonrepeated displays (which, due to their random variation, have little weight in determining the strategy). Finally, based on evolutionary considerations, learning the overall statistics of a scene environment would not only be more efficient, but also be more robust to environmental changes (in old scenes as well as the addition of new repeated scenes, which could be more easily incorporated in an environment-generic strategy) compared to learning highly specific display-target configurations.

Consistent with these functional considerations, there is also evidence from an fMRI study (including eye movements), by Manelis and Reder[44], in line with a procedural-learning account

of contextual cueing: When comparing the first with the last epoch of the experiment, Manelis and Reder[44] found a significant decrease in functional connectivity between hippocampus and sensory-procedural areas (which they did, however, not expressly attribute to mechanisms related to procedural oculomotor learning). Of note, hippocampal activity is not only involved in motor-sequence learning (e.g., in finger-tapping tasks[45]), but also in other statistical learning tasks, such as serial RT tasks (e.g., ref. [46,47],), particularly in the initial stages, with a decrease in activity later on[48]. This is consistent with a critical, but (over the course of practice) diminishing role of the hippocampus in procedural motor learning and would go some way to explain why patients with damage to the hippocampus do not display a contextual-cueing benefit[49]. Thus, a procedural-learning account involving the hippocampus could explain the phenomenon of contextual cueing without assuming the acquisition of display-specific contextual representations.

Such a display-generic, procedural account of oculomotor contextual cueing would also explain why participants usually do not explicitly recognize repeatedly encountered display arrangements:[1] hippocampal involvement would revolve around procedural aspects of search performance, rather than explicit (episodic) memory. This view is not necessarily incompatible with existing functional accounts of long-term memory, according to which hippocampus contributes to the formation of inter-element associations: for the present investigation, these associations would involve the binding of individual eye fixation – thus making procedural memory a specific instance of a more general, associative hippocampal memory system (e.g., ref. [49,50],).

**Limitations**. While our results suggest that display-unspecific, procedural learning of saccadic trajectories plays an important role in oculomotor contextual facilitation, it is important to note that if oculomotor scanning is allowed or encouraged, these developing strategies might look different from conditions in which observers are instructed to search the display without eye movements (e.g., ref. [51],). In line with this, electrophysiological (EEG) studies report evidence of display-specific contextual cueing – more precisely: target-side-specific lateralizations of event-related potentials indicative of attentional resource allocation – as early as around 200 ms post display onset[52] (see also ref. [53]). Arguably, disallowing eye movements would impede the evolution of generic scanning procedures, in particular, when brief exposure times prevent extended search. In contrast, more natural scenarios that require/allow oculomotor scanning foster the acquisition of display-unspecific routines adapted to the statistical regularities in the set of search displays encountered, which is dominated by repeated display arrangements. A somewhat related idea is that participants may have acquired specific memory representations of the display arrangement, but do not use them when they can scan the display freely[54]. The ERP task design, by contrast, likely forces display-specific learning, perhaps due to the need of holding individual display arrangements in working memory in order to solve the task[55]. Thus, there might be dual mechanisms underlying contextual facilitation, with the relative dominance of the display-generic over the display-specific mechanism being determined by the task demands, that is, the extent to which eye movements are possible/required or discouraged/dis-allowed.

## Conclusions

Contextual cueing is an important predictive-coding mechanism characterized by facilitation of search performance in repeated search arrays. As such, this facilitatory effect can be accommodated equally by accounts assuming associative learning of individual target locations in individual distractor arrangements (and the reproduction of individual scanpaths for these arrays) or, respectively, the acquisition of generic oculomotor patterns that optimize the scanning of the (for the participants initially overall new) set of search displays. The current study was designed to test the latter (display-generic) against the former (display-specific) account, by systematically investigating participants' eye-movement trajectories in repeated displays and comparing them against nonrepeated displays. Our findings are in line with a display-general scanning account of contextual facilitation: over time on task, scanpath sequences became increasingly similar across participants and displays, with total scanpath similarity being higher for repeated displays. We propose that at least under natural search conditions, contextual facilitation largely or exclusively derives from the acquisition of procedural oculomotor scanning programs, which become operational quite early during a given search trial. Conceptualizing contextual cueing as procedurally optimized oculomotor trajectories also offers new ways (1) for understanding the difficulty to update established contextual memories following consistent target-position changes within otherwise unchanged distractor arrangements (ref. [1], Experiment 3, ref. [7,56,57]); as well as (2) for understanding the apparent high capacity of contextual memory (see, e.g., ref. [58]; after being presented with 12 different target-distractor arrangements per day over a 5-day period, Jiang et al.[58].'s participants showed contextual facilitation for the total number of 60 arrangements when tested at the end); and (3), it provides a possible explanation as to why contextual cueing leads to fMRI activations in sensory brain areas contributing to procedural learning (e.g., ref. [44]; see also ref. [59], for confirmatory evidence using MEG). Moreover, a display-generic account of contextual learning would provide a coherent and, in terms of Occam's razor, the simplest explanation: it explains the search advantage for repeated versus nonrepeated displays, as well as the practice-dependent improvement of search in novel displays[1] in terms of procedural learning or skill acquisition[18]. Finally, given our evidence that oculomotor search is optimized independently of a particular configuration as a skill of performing a visual search task in general, we propose to use the more neutral term contextual facilitation (instead of display-specific contextual cueing) to describe the effects of procedural spatial learning in visual search. Repeated displays merely have a stronger influence in the optimization process, bringing about the facilitation effect.

## Data availability

The raw data that support the analysis and results are publicly accessible at https://osf.io/snjpk/.

## Code availability

The analysis code is available at https://osf.io/snjpk/.

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

## Acknowledgements

This research was supported by grants of the German Research Council (DFG) to Thomas Geyer (GE 1889/4-2 & RTG 2175). The funder had no role in study design, data collection and analysis, decision to publish or preparation of the manuscript.

## Author contributions

WS and AZ conceived and designed the experiments; AZ performed the experiments; WS analyzed the data; HJM contributed to conceptual analysis and discussion; WS, HJM, TG wrote the paper.

## Funding

## Competing interests

The authors declare no competing interests.
