## [Peer Review File · Communications Psychology]

16th Mar 23

Dear Mx Seitz,

Thank you for your patience during the peer-review process. Your manuscript titled "Statistical learning in visual search: 'contextual cueing' reflects the acquisition of an optimal, 'one-for-all' oculomotor scanning strategy" has now been seen by 2 reviewers, and I include their comments at the end of this message. They find your work of interest, but raised some important points. We are interested in the possibility of publishing your study in Communications Psychology, but would like to consider your responses to these concerns and assess a revised manuscript before we make a final decision on publication.

We therefore invite you to revise and resubmit your manuscript, along with a point-by-point response to the reviewers. Please highlight all changes in the manuscript text file.

Editorially, we ask you to prioritize the following main issues in revision.

- 1) Reviewer #1, and to a lesser degree Reviewer #2 questions, aspects of the analysis that presently do not preclude alternative accounts of the data. These issues need to be addressed through further analysis, or if this is not feasible on the present dataset, additional empirical data.
- 2) Especially Reviewer #2, but also Reviewer #1 highlights presentational issues. We ask you to clarify how exactly the study confirms, extends, or contradicts existing accounts, without overstating its implications, and avoiding any novelty claims (describing the study or results as "novel", "first", "unprecedented", etc).
- 3) To meet our requirements for reporting and interpreting statistics (<https://www.nature.com/commspsychol/submit/submission-guidelines#statistical-guidelines>), you must support all claims about null-effects with positive evidence for the null derived from Bayesian statistics or two-sided tests of equivalence. Please bear in mind these requirements as you revise the paper, including when you address Reviewer #2's concerns about the reporting of results and interpretation of the non-significant interaction contrast.
- 4) Your manuscript is lacking details on the study's ethics agreement and informed consent provided by the participants. Please ensure transparent reporting on both accounts.

We also recommend following the Reviewer's comment about substituting "efficient" for "optimal".

Please use the following link to submit your revised manuscript, point-by-point response to the referees' comments (which should be in a separate document to any cover letter) and the completed checklist:

[link redacted]

** This url links to your confidential home page and associated information about manuscripts you may have submitted or be reviewing for us. If you wish to forward this email to co-authors, please

delete the link to your homepage first **

Please do not hesitate to contact me if you have any questions or would like to discuss these revisions further. We look forward to seeing the revised manuscript and thank you for the opportunity to review your work.

Best regards,

Antonia Eisenkoeck

Antonia Eisenkoeck
Senior Editor
Communications Psychology

EDITORIAL POLICIES AND FORMATTING

Editorial Policy: [Policy requirements](https://www.nature.com/documents/nr-editorial-policy-checklist.pdf) (Download the link to your computer as a PDF.)

Furthermore, please align your manuscript with our format requirements, which are summarized on the following checklist:

[Communications Psychology formatting checklist](https://www.nature.com/documents/commsj-psychol-style-formatting-checklist-article.pdf)

and also in our style and formatting guide [Communications Psychology formatting guide](https://www.nature.com/documents/commsj-psychol-style-formatting-guide-accept.pdf) .

* **CODE AVAILABILITY:** All Communications Psychology manuscripts must include a section titled "Code Availability" at the end of the methods section. In the event of publication, we require that the custom analysis code supporting your conclusions is made available in a publicly accessible repository; at publication, we ask you to choose a repository that provides a DOI for the code; the link to the repository and the DOI will need to be included in the Code Availability statement. Publication as Supplementary Information will not suffice. We ask you to prepare code at this stage, to avoid delays later on in the process.

* **DATA AVAILABILITY:**

All Communications Psychology manuscripts must include a section titled "Data Availability" at the end of the Methods section or main text (if no Methods). More information on this policy, is available at <http://www.nature.com/authors/policies/data/data-availability-statements-data-citations.pdf>.

At a minimum the Data availability statement must explain how the data can be obtained and whether there are any restrictions on data sharing. Communications Psychology strongly endorses open sharing of data. If you do make your data openly available, please include in the statement:

We recommend submitting the data to discipline-specific, community-recognized repositories, where possible and a list of recommended repositories is provided at <http://www.nature.com/sdata/policies/repositories>.

If a community resource is unavailable, data can be submitted to generalist repositories such as [figshare](https://figshare.com/) or [Dryad Digital Repository](http://datadryad.org/). Please provide a unique identifier for the data (for example a DOI or a permanent URL) in the data availability statement, if possible. If the repository does not provide identifiers, we encourage authors to supply the search terms that will return the data. For data that have been obtained from publicly available sources, please provide a URL and the specific data product name in the data availability statement. Data with a DOI should be further cited in the methods reference section.

REVIEWERS' EXPERTISE:

Reviewer #1: contextual cueing, visual attention, statistical learning

Reviewer #2: visual attention, eye tracking

REVIEWERS' COMMENTS:

Reviewer #1 (Remarks to the Author):

The authors investigate the causes for the facilitation of visual search with repeated target-distractor displays. Previous research showed that search times decreased when the same spatial configuration of target-distractor displays is shown even though the actual target and distractor shapes change. The presumed reason was that participants established an implicit template of the repeated target-distractor configuration, which guides search towards the target location. The authors challenge this idea by suggesting that the benefit is due to more efficient general search strategies. Search becomes more efficient in general and because the repeated displays are searched more often, the optimization reflects these trials more strongly, which allows for faster search on repeated compared to novel displays.

The manuscript is well written and presents a novel approach to a classic problem. The proposed alternative account challenges conceived ideas with new methods. While I think that the idea proposed by the authors is convincing, it is also somewhat contradictory.

The problem is the following. The data shows that scan paths become more similar over time, but this can only be true for the initial portions of the trajectory. Just before finding the target, trajectories must deviate because the targets are in different quadrants. Thus, more efficient search predicts that the final fixations before locating the target become less similar over time because the participants move to the target location more efficiently. Maybe the impact of the final fixations could be evaluated by re-running the analysis of similarity on the initial 50% and final 50% of fixations. I would expect similarity to increase for initial fixation, but not for the final fixations. That is, the dissimilarity between target locations should be reflected in decreasing or unchanging similarity.

From a methodological point of view, the authors introduce new measures, and as a reader, I would like to know whether these measures reflect basic characteristics of the search task. The authors would need to show that increasing similarity is not an artefact of their procedure by showing that similarity decreases for parts of the trajectory where this appears trivial. That is, the endpoints are different, so similarity should remain unchanged or even decrease.

Table 1: To facilitate comparison across studies, it should be indicated how many pixels make up a degree of visual angle.

Lines 388-392: As the false-alarm rate is directly dependent on the hit rate (false-alarm = 1-hit-rate), a t-test between the two is not appropriate. However, one of these values could be compared to chance (.5) or d' prime could be calculated.

Line 489: missing word

Figure 1G) Top-right graph should be "Whitin-participants" not "Within-display"

It does not make much sense to report two decimals for latencies. Rounding to the next millisecond would remove some clutter without losing information. Also, p-values could be capped at .001 or .01 to avoid overestimation of the precision of these values.

Reviewer #2 (Remarks to the Author):

The study aims to examine whether contextual cueing depends on learning of specific, repeated display arrangements or on more generic learning of the whole set of displays, which would lead to procedural knowledge and enable oculomotor strategies which become overall more efficient during search. While the first view is presented as the traditional explanation of contextual cueing, the second view is presented as a new theoretical account of the phenomenon. The key used by the authors to disentangle between the two accounts is to introduce the analysis of scanpath similarity, with different measures, across displays for the same viewer and across viewers from the same display.

Examining whether observers are guided during search by specific memory templates for specific displays is an interesting question. Using scanpath measures to study this question is appropriate, and is useful methodological improvement compared to typical eye-movement measures used in contextual cueing research.

MAIN POINTS:

- 1) However, my main issue with this work is that, to the best of my understanding, its novelty is claimed in a stronger way than what it actually is. As the authors acknowledged in the Introduction, effects of task familiarisation and, thus, procedural learning have been acknowledged by several studies in the field, so the idea per se of a generic mechanism at play is not new. Thus, is the novelty of the present study mainly in the “weighting” of the contribution of display-specific learning vs. general improvement of scanning strategies to contextual cueing? Also, as specific and generic mechanisms can co-exist, they are not in opposition as the paper might sometime lead the reader to believe. Finally, I don’t think statistical learning in contextual cueing has always been considered in terms of the development of such highly specific guiding templates, as it would appear from how this paper discusses the literature.
- 2) Predictions: Does prediction 1, lines 145-147, also apply to the specific template account? If yes, this should be acknowledged. Moreover, prediction 2, about whether similarity measures should increase or not in repeated display is clear and I do agree it disentangles between different underlying mechanisms. However, shouldn’t this prediction depend on how different between them the repeated displays are? Have authors considered that?
- 3) If the target can appear only in one out of four locations in non-repeated display, is it not possible that the participants learned those specific locations and that, therefore, the overall results are also driven by this aspect? I’d like the authors to comment on that. I also would like the authors to consider more broadly the possible impact of the low variability of target positions (for both repeated and non-repeated displays) and the low variability across display overall in their study.
- 4) Would it be interesting to analyse first saccade direction and/or first saccade gain as a measure of learning? Would it help to disentangle the two accounts and to examine whether participants learned the possible few target positions even in non-repeated displays?
- 5) It has been shown (see e.g. the work of Vö, Wolfe et al.) that participants may have specific representations of the searched arrangement, but they do not use them when they can freely scan the display. I’d like the authors to explicitly comment on this in the Discussion.
- 6) Please specify the random structure of the models (did you include only the intercept or also the

slopes?). Why were Target Quadrant and Participant not included as random factors within the same model? Including both factors and also their slopes in the models (if the models converge) would allow for better taking into account the variability of the data due to these factors, and thus to obtain “cleaner” results. Moreover, I am not sure why you conducted ANOVAs and not mixed-model logistic regressions.

- 7) The terms “optimization”, “optimal”, etc. are used throughout, but wouldn’t be enough to refer to efficiency? At lines 539-540, the authors state: “These scanning strategies are optimal in the sense that they save processing energy”, for which “efficient” would then sound more appropriate. Moreover, for instance, if after many trials there still is an improvement of performance, as there seem to be in the present results, then there still is room for some enhancement of efficiency and, thus, optimality has not been reached. Whether viewer can reach truly optimal behaviour and whether reaching optimality is an aim to which the visual and cognitive systems should tend is a matter for further discussions, which I am not requiring the authors to undertake in their paper.

- 8) The clarity and readability of the paper would very much benefit from shorter sentences and fewer brackets throughout. As examples of a very long sentence, difficult to read, see lines 124-130 or 605-609.

MINOR POINTS:

- The Introduction contains some redundancies; for instance, lines 105-106 and 109-112 repeat concepts considered earlier on. Streamlining this section would enhance clarity and improve the flow of argument.

- Participant sub-section, justification of sample size: did previous studies used as reference also analyse the effects at the level of single displays? If not, how can the authors be confident that their “relatively large sample size” is enough to have “high statistical power” (line 187) in the present study? Even only consider the Block variable, it has 32 levels, and this may be challenging for the power of the study. Did this affect your possibility of finding significant interactions?

- Trial order: some confusion might arise between lines 171-172 and 211-213.

- Please provide more information (e.g., max error; tracked eye; how many points) about the calibration procedure.

- Line 347: I think you used the label “mixed-effect ANOVAs” as you used the lme4 package in R for mixed-effect modelling. However, this label might generate some confusion in the reader when used in the context of a fully repeated-measure design.

- Please report effect sizes.

- I’d advise to remove the decimals for the reported ms.

- The text in both x- and y-axes in Figure 1 is very small, difficult to read. Moreover, the examples in C) and D) are about targets in two different quadrants. It would be clearer to use examples about the same quadrant.

- Discussion, lines 466-467: "We also found the saccade amplitudes to be significantly shorter for repeated displays (in contrast to Tseng & Li, 2004)." Any explanation for this discrepancy?

- Discussion, lines 473-474, it is stated that scanpath measures "characterize the memory mechanism that drives the gains in the aggregate eye-movement indices". While I agree that they are useful measures, I found the claim too bold and/or generic. It would be useful to specify what "memory mechanism" the authors are referring to.

- Discussion, lines 480-481: "[...] practice-dependent gains were more strongly driven by displays sampled repeatedly (vs. displays sampled only once) over the course of the task." This statement could mislead the reader to think that the interaction between the type of display and the block was significant. Moreover, I think that the absence of a significant interaction should be discussed. Wouldn't the generic, "procedural-optimisation" account predict a significant interaction?

- Discussion, lines 488-489: incomplete sentence.

- Discussion, lines 516-517: "[...] to search the display covertly/passively (e.g., Lleras & von Mühlenen, 2004) or even without eye movements." This should be rephrased as covertly = without eye movements.

- Discussion, lines 528-529: "[...] assuming that 700 ms are not sufficient for full processing of the items, as indexed by mean RTs being typically in excess of 1000 ms in T vs. L-type letter search tasks" sounds like a rather weak speculation, as RTs also incorporate motor response time, not only perceptual processing.

- Discussion, line 547: I'd suggest avoiding the term "scene" to refer to these rather simple item arrays.

- Discussion, line 572: "[...] in the current context, these associations would involve the binding of individual eye fixations". What do the authors mean here?

-Appendix, caption of Figure 2: "As can be seen, scanpath lines representing individual participants are more similar for repeated as compared to non-repeated displays." This doesn't appear so obvious to me.

- Some measures, in particular mean saccade amplitude and saccade amplitude SD, look very variable across blocks. Moreover, the discrete Fréchet distance looks quite overlapping between repeated and non-repeated displays. Could the authors comment on these aspects?

June 8, 2023

Nature Communications Psychology

First revision of manuscript COMMSPSYCHOL-23-0031, “Statistical learning in visual search: ‘contextual cueing’ reflects the acquisition of an optimal, ‘one-for-all’ oculomotor scanning strategy”, by W. Seitz, A. Zinchenko, H. J. Müller, & T. Geyer

Dear Reviewers,

We would like to thank you for your valuable comments. All in all, we hope that we were able to address all the comments in a satisfactory manner, as we believe that the points raised were helpful in order to elevate the quality of our manuscript. In our revision, we have attempted to address all points made. How we have done this is outlined in the text below, together with the original recommendations. In addition, all changes have been marked in blue in the manuscript.

Amendments and Original Review

Reviewer #1

The authors investigate the causes for the facilitation of visual search with repeated target-distractor displays. Previous research showed that search times decreased when the same spatial configuration of target-distractor displays is shown even though the actual target and distractor shapes change. The presumed reason was that participants established an implicit template of the repeated target-distractor configuration, which guides search towards the target location. The authors challenge this idea by suggesting that the benefit is due to more efficient general search strategies. Search becomes more efficient in general and because the repeated displays are searched more often, the optimization reflects these trials more strongly, which allows for faster search on repeated compared to novel displays.

The manuscript is well written and presents a novel approach to a classic problem. The proposed alternative account challenges conceived ideas with new methods. While I think that the idea proposed by the authors is convincing, it is also somewhat contradictory.

Response. We would like to thank you for the overall positive evaluation of our manuscript.

The problem is the following. The data shows that scan paths become more similar over time, but this can only be true for the initial portions of the trajectory. Just before finding the target, trajectories must deviate because the targets are in different quadrants. Thus, more efficient search predicts that the final fixations before locating the target become less similar over time because the participants move to the target location more efficiently. Maybe the impact of the final fixations could be evaluated by re-running the analysis of similarity on the initial 50% and final 50% of fixations. I would expect similarity to increase for initial fixation, but not for the final fixations. That is, the dissimilarity between target locations should be reflected in decreasing or unchanging similarity.

From a methodological point of view, the authors introduce new measures, and as a reader, I would like to know whether these measures reflect basic characteristics of the search task. The authors would need to show that increasing similarity is not an artefact of their procedure by showing that similarity decreases for parts of the trajectory where this appears trivial. That is, the endpoints are different, so similarity should remain unchanged or even decrease.

Response. We entirely agree with this reasoning and the recommendations, which we have addressed them in three ways in a new paragraph titled “Validity check” (see pp. 18-20). First, we analyzed scanpath similarity separately for the ‘initial’ 50% of the fixations constituting the scanpaths and for the ‘final’ 50%, as suggested by the reviewer. We found that similarity was indeed significantly decreased for the ‘final’ part of the scanpaths. Second, we analyzed the physical distance of each fixation from the target location in the ‘initial’ and ‘final’ scanpath parts. This analysis revealed fixation-target distances to be substantially lower in the ‘final’ part. Third, and importantly, we also found that the ‘theoretical’ statistical scanpath-similarity measure(s) and the ‘empirical’ fixation-target distance measure correlated. Thus, our revised analyses, considering both a theoretical and empirical measure, indicate that the similarity of eye-movement trajectories reflects basic characteristics of the search: monotonicity of oculomotor scanpaths toward (in the various displays) individual-unique target regions in later phases of the search (e.g., Tseng & Li, 2004; *AP&P*; Pollmann & Manginelli, 2009; *Psych Res*). Accordingly, our finding of higher scanpath consistency for the set of repeated vs. non-repeated displays likely reflects common mechanisms determining the spatio-temporal scanning pattern in the early parts of ‘inefficient’ search.

Table 1: To facilitate comparison across studies, it should be indicated how many pixels make up a degree of visual angle.

Response. We added this info to the caption of Figure 1 (on p. 21).

Lines 388-392: As the false-alarm rate is directly dependent on the hit rate (false-alarm = 1-hit-rate), a t-test between the two is not appropriate. However, one of these values could be compared to chance (.5) or d'prime could be calculated.

Response. We thank the reviewer for alerting us to this issue. To clarify: in our analysis of recognition performance, we strictly followed previous attempts (e.g., Chun & Jiang, 1998; *Vis Cog*) and compared a rating score from the set of repeated displays (=hit rate: correctly identifying a repeated display as 'old') with a rating score coming from the set of non-repeated arrays (=false alarm rate: incorrectly judging a non-repeated display as 'old'). This means that both measures were obtained from different samples of either repeated or non-repeated displays and were thus independent. We nevertheless ran the alternative analysis suggested by the reviewer and compared the hit rates against the 50% baseline – i.e., chance – performance, which also yielded a non-significant result: $t(45) = 0.98$, $p = 0.33$, $BF = 0.25$ (see p. 23).

Line 489: missing word

Response. Corrected.

Figure 1G) Top-right graph should be “Whitin-participants” not “Within-display”
It does not make much sense to report two decimals for latencies.

Response. Corrected.

Rounding to the next millisecond would remove some clutter without losing information.

Response. Done.

Also, p-values could be capped at .001 or .01 to avoid overestimation of the precision of these values.

Response. We thank the reviewer for this suggestion, which we have implemented by capping p-values < 0.001 at 0.001 and reporting exact p-values otherwise.

Reviewer #2 (Remarks to the Author):

The study aims to examine whether contextual cueing depends on learning of specific, repeated display arrangements or on more generic learning of the whole set of displays, which would lead to procedural knowledge and enable oculomotor strategies which become overall more efficient during search. While the first view is presented as the traditional explanation of contextual cueing, the second view is presented as a new theoretical account of the phenomenon. The key used by the authors to disentangle between the two accounts is to introduce the analysis of scanpath similarity, with different measures, across displays for the same viewer and across viewers from the same display.

Examining whether observers are guided during search by specific memory templates for specific displays is an interesting question. Using scanpath measures to study this question is appropriate, and is useful methodological improvement compared to typical eye-movement measures used in contextual cueing research.

Response. We thank the reviewer for his/her overall rather favorable evaluation.

MAIN POINTS:

- 1) However, my main issue with this work is that, to the best of my understanding, its novelty is claimed in a stronger way than what it actually is. As the authors acknowledged in the Introduction, effects of task familiarisation and, thus, procedural learning have been acknowledged by several studies in the field, so the idea per se of a generic mechanism at play is not new. Thus, is the novelty of the present study mainly in the “weighting” of the contribution of display-specific learning vs. general improvement of scanning strategies to contextual cueing? Also, as specific and generic mechanisms can co-exist, they are not in opposition as the paper might sometime lead the reader to believe. Finally, I don’t think statistical learning in contextual cueing has always been considered in terms of the development of such highly specific guiding templates, as it would appear from how this paper discusses the literature.

RESPONSE: We acknowledge that our exposition in the original version of the ms may have invited misunderstanding, which helped us clarify the questions at issue (see pp. 6-7 of the revision).

Indeed, that procedural learning plays a role in contextual-cueing experiments (as in all search tasks) has been widely acknowledged (including in our own studies) – in order to explain the *general* practice-related gains in search-RT speed over the course of the experiment. These general gains are assumed to be ‘measured’ by the speed-up of the RTs for non-repeated displays, which, by design, do not afford

contextual learning; it is assumed that these general gains are *the same* for repeated displays. The latter, however, show additional gains (over and above the speed-up for non-repeated displays) – which are attributed to contextual learning of the target-distractor relations in repeated displays. So, the standard account(s) assume that the general procedural and the specific contextual learning effects reflect *separate* learning mechanisms (as we have tried to explain on p. 6). Our new account, by contrast, is a *unitary* account, in that it explains both the general and the specific gains in terms of the procedural tuning (optimization) of the oculomotor scanning strategy to the regularities prevailing in the whole set of displays that observers encounter over the task. Since repeated displays are, by definition, encountered frequently and non-repeated displays only once, the former accrue greater weight in shaping the scanning strategy. We believe this is a novel idea that, to the best of our knowledge, has not been set out in this way in the extant literature. Of note also, existing *computationally explicit* models of contextual cueing (e.g., Brady & Chun, 2007; *JEP:HPP*; Beesley et al., 2016; *JEP:LMC*) are all variants of the ‘display-specific template’ type and (designed to) explain only the contextual-facilitation effect, ignoring the general procedural task-learning gains. There is as yet no computational model that explains both effects unitarily in terms of procedural optimization.

We agree that, in principle, both mechanisms – procedural optimization and the acquisition of display-specific search-guiding ‘templates’ might be at work, in tandem, in contextual-cueing paradigms. That is, contextual facilitation may involve *both* the acquisition of contextual LTM representations that are based on associations between a repeated spatial context with a given invariant target location (i.e., template-based guidance in the present ms and introduced in Chun and Jiang’s CC pioneering/anchoring study as “[display specific] instances stored in memory”; e.g., Chun & Jiang, 1998, *Cogn Psychol*, p. 62) and procedural long-term learning of how to optimally scan the ‘world’ (or set) of search displays presented in a given acquired study of contextual cueing. However, the latter type of contextual learning may be more important for, or even the sole determinant of, the facilitation effect in search scenarios that require serial inspection of the displays by the eyes to accomplish the task.

Given that the latter type of learning has been relatively neglected (at least in the few computationally explicit models of contextual cueing; but also in empirical studies that considered only aggregate measures across the sets of repeated and, respectively, non-repeated displays, rather than complex scanpath measures), we focus on – and in the previous version perhaps over-emphasized – the latter mechanism (examining scanpath measures in addition to aggregate measures). We now acknowledge that display-specific template-based guidance and display-general procedural optimization of search are not mutually exclusive and may be at

work in tandem. Our changes to the Introduction (on pp. 6-7) also allowed us to connect the Introduction more tightly with the Discussion section, where we consider the (relative) interplay of the alternative CC mechanisms in greater detail (pp. 30-31).

- 2) Predictions: Does prediction 1, lines 145-147, also apply to the specific template account? If yes, this should be acknowledged. Moreover, prediction 2, about whether similarity measures should increase or not in repeated display is clear and I do agree it disentangles between different underlying mechanisms. However, shouldn't this prediction depend on how different between them the repeated displays are? Have authors considered that?

Response. We now acknowledge that prediction 1 also applies to the display-specific account (p. 8). Regarding prediction 2: we agree that this prediction would depend on the similarity of the repeated displays themselves. Of note, though, in our display design, we carefully implemented measures that rendered the repeated displays as *dissimilar* as possible. We emphasize this in several places of the revision. Examples: we deliberately located our targets in the repeated (as well as in the non-repeated) displays in different display quadrants (see p. 9). Further, our search displays had overall 40 possible locations (4, 8, 12, and 16 locations on ring 1, 2, 3, and 4, respectively). So, when randomly placing the 12 search items on this grid (the only constraint being that each quadrant contains an equal number of 3 items), there are, in principle, more than $2.07 \cdot 10^9$ possible configurations of individual-unique search-displays (p. 12). Selecting our repeated (and non-repeated) displays randomly from this large space of possible configurations rendered it quite unlikely that they were structurally highly similar. Given this, we believe scanpath-similarity findings are not just an artifact of the way we generated our displays. In support of this, (responding to a point made by reviewer #1) we have now also analyzed scanpath similarity separately for the first and second halves of fixations (i.e., the 'initial' 50% and the 'final' 50%) making up the scanpath on a trial. As expected, we found (and now also report) that similarity is decreased for the 'final' scanpath parts, as the eyes approach the individual-unique target regions in the later phases of the search (see new paragraph "Validity check" on pp. 18-20). Also, in our linear mixed models (pp. 14-15), we explicitly consider/correct for random variability coming from individual displays (and participants) in order to render our results as generalizable as possible to other displays/participants.

- 3) If the target can appear only in one out of four locations in non-repeated display, is it not possible that the participants learned those specific locations and that, therefore, the overall results are also driven by this aspect? I'd like the authors to comment on that. I also would like the authors to consider more broadly the

possible impact of the low variability of target positions (for both repeated and non-repeated displays) and the low variability across display overall in their study.

Response. We agree that these are important points, which we now explicitly discuss in the revision (see p. 10 and pp. 12-13). To briefly elaborate here: Fixing the absolute target locations (see p. 12) in non-repeated displays (while of course randomizing the distractor configuration) is standard practice in contextual-cueing studies to control for absolute ‘target-location probability-cueing’ effects (e.g., Geng & Behrmann, 2005; AP&P) when comparing search performance to repeated displays, in which the absolute target position is fixed by definition (as well as the distractor configuration). That is, it is a necessary design feature to isolate contextual facilitation, by eliminating any facilitation that may derive from learning the absolute target positions. Concerning your other points, which we addressed by new analysis: we find a relative to previous studies of contextual cueing comparable contextual-facilitation effect-size score in our study (p. 10), making it unlikely that our (2 x 4) target positions in repeated and non-repeated arrays impacted our results. To rule display-specific effects, we consider target quadrant/distractor display as random factor in our mixed-effects models (see pp. 14-15); and our responses to your previous comment #2).

- 4) Would it be interesting to analyse first saccade direction and/or first saccade gain as a measure of learning? Would it help to disentangle the two accounts and to examine whether participants learned the possible few target positions even in non-repeated displays?

Response. We thank the reviewer for alerting us to this possibility, which we examined by additional linear models with block and context as fixed factors and participant and target quadrant as random factors. The results revealed no difference in the mean distance of the first fixation from the target between repeated and non-repeated displays ($p > 0.3$); the 2-way context x block interaction was also not significant, $p > 0.75$, BFs for main and interaction effect $< 10^{-56}$). This suggests at least that if absolute target-position learning was at work, it would not have differed between non-repeated and repeated displays. Given the null effect, we refrain from reporting this analysis in the revision – though of course we would be happy to include it if wished by the reviewer.

- 5) It has been shown (see e.g. the work of Vö, Wolfe et al.) that participants may have specific representations of the searched arrangement, but they do not use them when they can freely scan the display. I’d like the authors to explicitly comment on this in the Discussion.

Response. This is an interesting suggestion, which made us re-think the motivation of our study and the interpretation of the data (see pp. 6-7 of the Introduction and

pp. 30-31 of the Discussion section). – We now acknowledge the possible co-existence of display-specific and display-generic mechanisms in driving contextual facilitation, with the relative dominance of one over the other mechanism being dependent on whether oculomotor scanning is required/encouraged or whether conditions allow the task to be performed without eye movements (see also our response to your comment #1).

- 6) Please specify the random structure of the models (did you include only the intercept or also the slopes?). Why were Target Quadrant and Participant not included as random factors within the same model? Including both factors and also their slopes in the models (if the models converge) would allow for better taking into account the variability of the data due to these factors, and thus to obtain “cleaner” results. Moreover, I am not sure why you conducted ANOVAs and not mixed-model logistic regressions.

Response. We thank the reviewer for this suggestion. We used linear random-intercept models (rather than logistic regressions as our dependent variables – of RTs, error rates, oculomotor measures – were numerical variables) to account for ‘unwanted’ variability deriving from individual displays (with different target locations/quadrants) and individual participants (assuming that RTs typically vary across individual participants and target locations/hemifields; see, e.g., Olson & Chun, 2002; *Vis Cog*).

- 7) The terms “optimization”, “optimal”, etc. are used throughout, but wouldn’t be enough to refer to efficiency? At lines 539-540, the authors state: “These scanning strategies are optimal in the sense that they save processing energy”, for which “efficient” would then sound more appropriate. Moreover, for instance, if after many trials there still is an improvement of performance, as there seem to be in the present results, then there still is room for some enhancement of efficiency and, thus, optimality has not been reached. Whether viewer can reach truly optimal behaviour and whether reaching optimality is an aim to which the visual and cognitive systems should tend is a matter for further discussions, which I am not requiring the authors to undertake in their paper.

Response. We thank the reviewer for raising this point, which led us to clarify our concept of ‘optimality’ (see Footnote 1 on p. 6). Please note that we did not substitute the term *optimality* – because we mean to set our approach apart from already existing work (i.e., Tseng & Li, 2004; *AP&P*) that explicitly uses the alternative term of ‘*efficient*’ (versus *inefficient*) to characterize the pattern of eye movements in search tasks. Moreover, our ms is already out as preprint with > 150

reads, so changing it – and specifically the title (also containing “optimal”) – would probably create difficulties with any preprint citations.

- 8) The clarity and readability of the paper would very much benefit from shorter sentences and fewer brackets throughout. As examples of a very long sentence, difficult to read, see lines 124-130 or 605-609.

Response. We changed this sentence and overall tried to avoid longer sentences in the paper. Examples are on p. 6, p.7, and on p. 34.

MINOR POINTS:

- The Introduction contains some redundancies; for instance, lines 105-106 and 109-112 repeat concepts considered earlier on. Streamlining this section would enhance clarity and improve the flow of argument.

Response. We deleted this duplicate in the introduction.

- Participant sub-section, justification of sample size: did previous studies used as reference also analyse the effects at the level of single displays? If not, how can the authors be confident that their “relatively large sample size” is enough to have “high statistical power” (line 187) in the present study? Even only consider the Block variable, it has 32 levels, and this may be challenging for the power of the study. Did this affect your possibility of finding significant interactions?

Response. We explicated our choice of sample size in the revision (pp. 10-11).

- Trial order: some confusion might arise between lines 171-172 and 211-213.

Response. We now removed this inconsistency in the description of our experimental design.

- Please provide more information (e.g., max error; tracked eye; how many points) about the calibration procedure.

Response. We added this information to the revision (on p. 13)

- Line 347: I think you used the label “mixed-effect ANOVAs” as you used the lme4 package in R for mixed-effect modelling. However, this label might generate some confusion in the reader when used in the context of a fully repeated-measure design.

Response. We clarified this by ‘just’ referring to our statistical approach as “linear mixed models” (p. 15) and consistently avoiding/deleting the then misleading terms “repeated-measures” or “ANOVA”.

- Please report effect sizes.

Response. We agree with this suggestion and now report (R2) effect-size measures (p. 15). Please note that we chose the R2 statistic in order to reveal the strength, or aptness, of our linear mixed models in predicting the dependent measures based on the entire set of independent, i.e., fixed, factors.

- I'd advise to remove the decimals for the reported ms.

Response: Thanks, we removed decimals in the revision.

- The text in both x- and y-axes in Figure 1 is very small, difficult to read. Moreover, the examples in C) and D) are about targets in two different quadrants. It would be clearer to use examples about the same quadrant.

Response: We improved the manuscript in this regard.

- Discussion, lines 466-467: "We also found the saccade amplitudes to be significantly shorter for repeated displays (in contrast to Tseng & Li, 2004)." Any explanation for this discrepancy?

Response. This finding of reduced saccade amplitudes in repeated displays is also somewhat unclear to us, though it is not unexpected given that a previous oculomotor study of contextual cueing (Tseng & Li, 2004; *AP&P*) also found at least numerically reduced saccade amplitudes in the repeated displays. In terms of a functional interpretation, the reduced saccade amplitudes may be diagnostic of a coarse-to-fine search strategy (e.g., Over et al., 2007, *Vis Res*): when encountering a new search stimulus, participants typically start by making rather long saccades, but saccadic amplitudes decrease as the search progresses. Applied to the present investigation, the reduction in saccadic amplitudes for repeated versus non-repeated arrays may then index stronger coarse-to-fine oculomotor scanning in repeated displays – which would be in line with the idea of a generic procedural mechanism receiving stronger tuning from the repeated displays. Please note that we do mention this idea in the revised ms (as it is post-hoc and does not go beyond our central new hypothesis – of procedural tuning of broad, i.e., display-generic, oculomotor scanning driven predominantly by the repeated displays).

- Discussion, lines 473-474, it is stated that scanpath measures "characterize the memory mechanism that drives the gains in the aggregate eye-movement indices". While I agree that they are useful measures, I found the claim too bold and/or generic. It would be useful to specify what "memory mechanism" the authors are referring to.

Response. We have rewritten the critical sentence on p. X: *“Having established comparability of the present findings with those reported in prior studies of contextual cueing, we went on to analyze oculomotor scanpaths in order to differentiate between a display-specific and a procedural, display-generic account of acquired contextual facilitation that may drive the gains in the aggregate eye-movement indices...”*

- Discussion, lines 480-481: “[...] practice-dependent gains were more strongly driven by displays sampled repeatedly (vs. displays sampled only once) over the course of the task.” This statement could mislead the reader to think that the interaction between the type of display and the block was significant. Moreover, I think that the absence of a significant interaction should be discussed. Wouldn't the generic, “procedural-optimisation” account predict a significant interaction?

Response. We agree with this suggestion because, indeed, the current statement is misleading. We have therefore clarified this sentence: *“... we found that, while the consistency of the scanpaths increased overall with increasing time on task for both repeated and non-repeated displays, these practice-dependent gains were more strongly driven by displays sampled repeatedly (vs. displays sampled only once)...”*. Concerning the absence of an interaction effect: we now report Bayesian statistics for inference (see also our response to your comment on sample size selection above), which effectively yielded only a low probability for the alternative hypothesis (of a context x block interaction; see, p. 25). Moreover, the absence of an interaction would also be expected from prior studies of contextual cueing showing that contextual cueing develops rather early during search and reaches a plateau even after the first few blocks of trials (see, e.g., the review article of Goujon et al., 2015; *TICS*).

- Discussion, lines 488-489: incomplete sentence.

Response. Corrected. Thank you!

- Discussion, lines 516-517: “[...] to search the display covertly/passively (e.g., Lleras & von Mühlennen, 2004) or even without eye movements.” This should be rephrased as covertly = without eye movements.

Response. Corrected.

- Discussion, lines 528-529: “[...] assuming that 700 ms are not sufficient for full processing of the items, as indexed by mean RTs being typically in excess of 1000

ms in T vs. L-type letter search tasks” sounds like a rather weak speculation, as RTs also incorporate motor response time, not only perceptual processing.

Response. We agree with this suggestion and have now substantially rewritten this paragraph (see pp. 30-31), removing speculations and embedding it in a stronger theoretical context that considers, e.g., the work of Ballard et al., (1995; *JOCN*) and Vö & Wolfe (2012; *JEP:LMC*) on the interplay of display-generic and display-specific learning in search tasks.

- Discussion, line 547: I’d suggest avoiding the term “scene” to refer to these rather simple item arrays.

Response. Corrected.

- Discussion, line 572: “[...] in the current context, these associations would involve the binding of individual eye fixations”. What do the authors mean here?

Response. We clarified this sentence: “*For the present investigation, these associations would involve the binding of individual eye fixations –...*”

-Appendix, caption of Figure 2: “As can be seen, scanpath lines representing individual participants are more similar for repeated as compared to non-repeated displays.”. This doesn’t appear so obvious to me.

Response. We have elaborated on the higher similarity of repeated over non-repeated displays in the description of the data shown in Figure 2 (on p. 41).

- Some measures, in particular mean saccade amplitude and saccade amplitude SD, look very variable across blocks. Moreover, the discrete Fréchet distance looks quite overlapping between repeated and non-repeated displays. Could the authors comment on these aspects?

Response. We confirm the reviewer’s observation that our standard error bars in the saccade amplitude/Frechét-distance measures do overlap, though the p value of the associated context effect is smaller than .05, likely due to these parameters representing different statistical properties of our data (of the uncertainty of the mean dependent on sample size vs. group differences in our dependent measure; for evidence of a parameter dissociation, see, e.g., Krzywinski & Altman, 2013; *Nat Methods*). Please note that, in an attempt to remove this inconsistency, we also tried the alternative variability measure of within-subject error bars, which are typically better suited to illustrate effects in a repeated-measures design. However, almost none of our participants displayed consistently high or low similarity values, so the benefits of this measure (which corrects for high vs. low baseline values of individual participants) were only marginal – which led us to keep the original standard error measure.

3rd Jul 23

Dear Mx Seitz,

Thank you again for the revisions of your manuscript and your patience during the re-review process. Your manuscript titled "Statistical learning in visual search: 'contextual cueing' reflects the acquisition of an optimal, 'one-for-all' oculomotor scanning strategy" has now been seen by the same 2 reviewers as before and I include their comments at the end of this message. The reviewers confirm that the work is much improved and we appreciate all the changes you made in response to the previous round of comments. At this stage, Reviewer #2 raised some final points which we ask you to address before making a final decision.

We therefore invite you to revise and resubmit your manuscript, along with a point-by-point response to the reviewers. Please highlight all changes in the manuscript text file.

To facilitate further processing of your work, I attach a checklist that sets out the formatting guidelines for manuscripts published in Communications Psychology. We ask you to pay close attention to each individual item. I also highlight on the checklist the issues that most often cause delays to acceptance of a manuscript.

We very strongly recommend that you already deposit the analysis code following the revisions, so that your code-availability statement is up to date and can be verified in the next instance.

Also, please update the Reporting Summary to mark that you conducted Bayesian analysis, add any additional information for the Bayesian analysis as well as effect sizes for all tests.

Please use the following link to submit your revised manuscript, point-by-point response to the referees' comments (which should be in a separate document to any cover letter) and the completed checklist:

[link redacted]

We hope to receive your revised paper within 4 weeks; please let us know if you aren't able to submit it within this time so that we can discuss how best to proceed. If we don't hear from you, and the revision process takes significantly longer, we may close your file. In this event, we will still be happy to reconsider your paper at a later date, provided it still presents a significant contribution to the literature at that stage.

Please do not hesitate to contact me if you have any questions or would like to discuss these revisions further. We look forward to seeing the revised manuscript and thank you for the opportunity to review your work.

Best regards,

Antonia Eisenkoeck

Antonia Eisenkoeck
Senior Editor
Communications Psychology

EDITORIAL POLICIES AND FORMATTING

Editorial Policy: [Policy requirements](https://www.nature.com/documents/nr-editorial-policy-checklist.pdf) (Download the link to your computer as a PDF.)

* **CODE AVAILABILITY:** All Communications Psychology manuscripts must include a section titled "Code Availability" at the end of the methods section. In the event of publication, we require that the custom analysis code supporting your conclusions is made available in a publicly accessible repository; at publication, we ask you to choose a repository that provides a DOI for the code; the link to the repository and the DOI will need to be included in the Code Availability statement. Publication as Supplementary Information will not suffice. We ask you to prepare code at this stage, to avoid delays later on in the process.

* **DATA AVAILABILITY:**

All Communications Psychology manuscripts must include a section titled "Data Availability" at the end of the Methods section or main text (if no Methods). More information on this policy, is available at <http://www.nature.com/authors/policies/data/data-availability-statements-data-citations.pdf>.

At a minimum the Data availability statement must explain how the data can be obtained and whether there are any restrictions on data sharing. Communications Psychology strongly endorses open sharing of data. If you do make your data openly available, please include in the statement:

- Unique identifiers (such as DOIs and hyperlinks for datasets in public repositories)
- Accession codes where appropriate

- If applicable, a statement regarding data available with restrictions
- If a dataset has a Digital Object Identifier (DOI) as its unique identifier, we strongly encourage including this in the Reference list and citing the dataset in the Data Availability Statement.

We recommend submitting the data to discipline-specific, community-recognized repositories, where possible and a list of recommended repositories is provided at <http://www.nature.com/sdata/policies/repositories>.

If a community resource is unavailable, data can be submitted to generalist repositories such as [figshare](https://figshare.com/) or [Dryad Digital Repository](http://datadryad.org/). Please provide a unique identifier for the data (for example a DOI or a permanent URL) in the data availability statement, if possible. If the repository does not provide identifiers, we encourage authors to supply the search terms that will return the data. For data that have been obtained from publicly available sources, please provide a URL and the specific data product name in the data availability statement. Data with a DOI should be further cited in the methods reference section.

REVIEWERS' COMMENTS:

Reviewer #1 (Remarks to the Author):

The authors have adequately addressed my previous concerns.
Line 481 : “targets positions” should be “target position”

Reviewer #2 (Remarks to the Author):

The authors did a very good and thorough job with the revision, making the paper theoretically and methodologically clearer and more impactful. Overall, I am satisfied with the responses to my comments.

However, before acceptance, I think that it would be important to include - perhaps in Supplementary Information - the specific structure of each model used: i.e. to present the specific random structure, including the slopes for the participant and item (display) terms, of the model that converged, and whose results are reported. If the full model was used and converged, please state that. If there were any convergence issues, so that models had to be simplified, it would be useful to state the simplification logic followed.

Did you remove any data due to poor calibration/validation?

Moreover, the highest mean calibration error accepted should be declared, besides the maximum

error in each of the points.

BFs: I agree that their inclusion is a good addition to the paper; they should also be included for non-significant results though, for instance top p. 22 (all $BF < \dots$).

Munich, July 12, 2023

Second revision of manuscript COMMSPSYCHOL-23-0031A

Dear Reviewers,

attached to this letter is the second revision of the manuscript “Statistical learning in visual search: ‘contextual cueing’ reflects the acquisition of an optimal, ‘one-for-all’ oculomotor scanning strategy”, which my co-authors and I would like to resubmit for possible publication in *Nature Communications Psychology*. We would like to thank the two expert reviewers for their positive evaluation of our work.

We have made every effort to revise the manuscript in accordance with your remaining comments, as I will describe in more detail below. Of course, we hope that you find this version ready for publication. Thanks again to you and for your helpful comments on the previous version.

Sincerely,
Werner Seitz
(on behalf of all co-authors)

Amendments and Original Review

Reviewer #1 (Remarks to the Author):

The authors have adequately addressed my previous concerns.

Line 481 : “targets positions” should be “target position”

Response. We thank Reviewer 1 for his/her overall positive assessment of our work. We have corrected the typo on p. 23.

Reviewer #2 (Remarks to the Author):

The authors did a very good and thorough job with the revision, making the paper theoretically and methodologically clearer and more impactful. Overall, I am satisfied with the responses to my comments.

Response. We thank Reviewer 2 for his/her encouraging feedback!

However, before acceptance, I think that it would be important to include - perhaps in Supplementary Information - the specific structure of each model used: i.e. to present the specific random structure, including the slopes for the participant and item (display) terms, of the model that converged, and whose results are reported. If the full model was used and converged, please state that. If there were any convergence issues, so that models had to be simplified, it would be useful to state the simplification logic followed.

Response. We agree with the reviewers’ view and now explicitly state (the R code of) our mixed-models in Appendix III (on p. 45).

Did you remove any data due to poor calibration/validation?

Response. We did not exclude any data due to poor calibration or validation, as we now make explicit on p. 13.

Moreover, the highest mean calibration error accepted should be declared, besides the maximum error in each of the points.

Response. We provide these values also on p. 13.

BFs: I agree that their inclusion is a good addition to the paper; they should also be included for non-significant results though, for instance top p. 22 (all $BF < \dots$).

Response. We now report BFs for the null findings on p. 22.

27th Jul 23

Dear Mx Seitz,

Your manuscript titled "Statistical learning in visual search: 'contextual cueing' reflects the acquisition of an optimal, 'one-for-all' oculomotor scanning strategy" has now been editorially evaluated. I am delighted to say that we are happy, in principle, to publish a suitably revised version in Communications Psychology under the open access CC BY license (Creative Commons Attribution v4.0 International License).

We therefore invite you to revise your paper one last time to address the remaining editorial requests. At the same time we ask that you edit your manuscript to comply with our format requirements and to maximise the accessibility and therefore the impact of your work.

EDITORIAL REQUESTS:

A key issue I want to draw your attention to are our data and code sharing requirements. The analysis code, and ideally also the experimental code should be deposited publicly; this deposition should preferably be in a DOI-minting repository that allows version control.

We really appreciate that you will share the raw data publicly. Please remember to deidentify raw data. In addition to the raw data deposition (which is very strongly encouraged but not obligatory), we mandate deposition of the numerical source data for the Figures. These data should be presented in the repository in a way that they can be immediately linked to the Figure.

More information on code and data policies is included in the attached editorial request table and accessible on our policy pages. Please note that Data availability statement and Code availability statement should be separate sections.

SUBMISSION INFORMATION:

OPEN ACCESS:

Communications Psychology is a fully open access journal. Articles are made freely accessible on publication under a <http://creativecommons.org/licenses/by/4.0> target="_blank"> CC BY

license (Creative Commons Attribution 4.0 International License). This license allows maximum dissemination and re-use of open access materials and is preferred by many research funding bodies.

For further information about article processing charges, open access funding, and advice and support from Nature Research, please visit <https://www.nature.com/commpsychol/article-processing-charges>

At acceptance, you will be provided with instructions for completing this CC BY license on behalf of all authors. This grants us the necessary permissions to publish your paper. Additionally, you will be asked to declare that all required third party permissions have been obtained, and to provide billing information in order to pay the article-processing charge (APC).

* TRANSPARENT PEER REVIEW: Communications Psychology uses a transparent peer review system. On author request, confidential information and data can be removed from the published reviewer reports and rebuttal letters prior to publication. If you are concerned about the release of confidential data, please let us know specifically what information you would like to have removed. Please note that we cannot incorporate redactions for any other reasons.

[link redacted]

Best regards,

Antonia Eisenkoeck

Antonia Eisenkoeck
Senior Editor
Communications Psychology